# GM-CSF drives dysregulated hematopoietic stem cell activity and pathogenic extramedullary myelopoiesis in experimental spondyloarthritis

Daniel Regan-Komito [1,4], James W. Swann [1,4], Philippos Demetriou[1], E. Suzanne Cohen[2], Nicole J. Horwood [1,3], Stephen N. Sansom [1] & Thibault Griseri [1]*

Dysregulated hematopoiesis occurs in several chronic inflammatory diseases, but it remains unclear how hematopoietic stem cells (HSCs) in the bone marrow (BM) sense peripheral inflammation and contribute to tissue damage in arthritis. Here, we show the HSC gene expression program is biased toward myelopoiesis and differentiation skewed toward granulocyte-monocyte progenitors (GMP) during joint and intestinal inflammation in experimental spondyloarthritis (SpA). GM-CSF-receptor is increased on HSCs and multi-potent progenitors, favoring a striking increase in myelopoiesis at the earliest hematopoietic stages. GMP accumulate in the BM in SpA and, unexpectedly, at extramedullary sites: in the inflamed joints and spleen. Furthermore, we show that GM-CSF promotes extramedullary myelopoiesis, tissue-toxic neutrophil accumulation in target organs, and GM-CSF prophylactic or therapeutic blockade substantially decreases SpA severity. Surprisingly, besides CD4$^+$ T cells and innate lymphoid cells, mast cells are a source of GM-CSF in this model, and its pathogenic production is promoted by the alarmin IL-33.

---

[1] Kennedy Institute of Rheumatology, Nuffield Department of Orthopaedics, Rheumatology and Musculoskeletal Sciences, University of Oxford, Roosevelt Drive, Oxford, UK. [2] Biopharmaceutical Research Division, AstraZeneca, Cambridge, UK. [3] Norwich Medical School, University of East Anglia, Bob Champion Research and Education Building, James Watson Road, Norwich Research Park, Norwich, UK. [4] These authors contributed equally: Daniel Regan-Komito, James W. Swann. *email: thibault.griseri@kennedy.ox.ac.uk

    1

Long-term hematopoietic stem cells (LT-HSC) differentiate through short-term HSC (ST-HSC) and then multipotent progenitor (MPP) stages into lineage restricted progenitors, i.e., lymphoid, myeloid, or megakaryocyte/erythroid progenitors[1]. Recently, dysregulation of hematopoietic stem and progenitor cell (HSPC) activity in the bone marrow (BM) has been reported in several chronic inflammatory diseases, including inflammatory bowel disease[2,3], lupus[4], and atherosclerosis[5,6]. However, the mechanism by which HSPCs inside the BM detect and respond to peripheral inflammation affecting remote target tissues remains elusive. Some clues have emerged from research into anti-microbial immunity[7], where studies have revealed that, although sheltered inside the BM niche, HSCs are more reactive to environmental cues than previously appreciated. Indeed, HSCs express receptors for inflammatory cytokines (e.g., receptors for IFN-α/β, IFN-γ, and IL-1β) and microbial compounds (e.g., TLR2 and TLR4), which upon activation trigger HSC proliferation and differentiation[7–9].

In this study, we investigate how HSC and downstream progenitors respond to the systemic inflammation characterizing spondyloarthritis (SpA). SpA encompasses several inflammatory arthropathies, including ankylosing spondylitis (AS), psoriatic arthritis, and enteropathic arthritis, which share common genetic and environmental triggers and pathophysiological mechanisms[10]. These include, for example, involvement of the IL-23/Th17 pathway and intestinal dysbiosis[11]. SpA is a systemic disease, such that inflammation affects not only the peripheral and axial joints but also other tissue sites, especially the barrier surfaces of the skin and intestine. Mechanistically, inflammation triggered by microbial stress at barrier surfaces together with mechanical stress in the joints of individuals with a predisposing genetic background are thought to contribute to SpA development[10,12].

At the cellular level, although the immune cell network that drives SpA is still unclear, genetic and experimental evidence supports a central role for innate cells in disease development[10,12]. For example, in response to microbial or sterile inflammatory stimuli, macrophages and dendritic cells produce excessive amounts of cytokines promoting arthritogenic Th17 cell response, e.g., IL-23, IL-6, and IL-1β. In addition, neutrophils accumulate in the synovial fluid of SpA patients and inflamed entheses (i.e., tendon and ligament insertions into the bone). These over-activated neutrophils are tissue-toxic owing to release of proteases, reactive oxygen species, and inflammatory cytokines that contribute to chronic bone and cartilage damage[13,14]. Importantly, neutrophils are short-lived cells and their peripheral numbers are highly dependent on the output from HSPCs in the BM[15].

Here, we ask how HSCs and myeloid progenitor cells in the BM respond to the inflammation targeting peripheral tissues in arthritis. We utilize a recently described model of SpA characterized by chronic arthritis development, enteritis, and systemic inflammation with numerous clinical features observed in AS patients[16]. In particular, we investigate how the chronic inflammation occurring in SpA influences the gene expression programs of HSCs and MPPs and impacts their differentiation into the various lineage-committed progenitors. Consequently, we highlight a key role for granulocyte-macrophage colony stimulating factor (GM-CSF) in regulating these processes in the BM and at extramedullary sites. In addition, we identify mast cells (MC) as a cellular source of GM-CSF during SpA and reveal IL-33 as a potent inducer of GM-CSF secretion, the latter promoting the production of tissue-toxic neutrophils in the BM and at extramedullary sites.

hypothesized that HSC activity in the BM may be modulated by signals that originate remotely in inflamed peripheral organs in SpA and travel to the BM via the bloodstream. To test this notion, we utilized arthritis-prone SKG mice[17] in the context of a novel model of SpA in which an IP injection of the microbial polysaccharide curdlan, a potent inducer of IL-23-mediated Th17 responses[18], triggers a SpA-like phenotype (Fig. 1a) replicating many features of AS in humans[16]. We confirmed these observations as curdlan-injected SKG mice developed a non-resolving inflammation characterized after ~1 month by severe paw swelling and draining lymph node enlargement, small intestinal inflammation, and enthesitis associated with ankle swelling and new bone formation. Conversely, SKG mice injected with phosphate-buffered saline (PBS) remained healthy (Fig. 1b, Supplementary Fig. 1a, b). In addition to the joint phenotype, we observed signs of systemic inflammation in spondyloarthritic SKG mice, manifesting as pale BM (a sign of anemia), splenomegaly (Fig. 1b), an intestinal increase in expression of inflammatory genes, i.e., Il1, Il6, Il23, and Tnf (Supplementary Fig. 1c), and weight loss (Supplementary Fig. 1a).

To identify HSCs and downstream progenitors, we utilized a well-defined panel of fluorescence-activated cell sorting (FACS) markers (Supplementary Fig. 1d)[19]. Compared to healthy controls, spondyloarthritic mice assessed ~6 weeks after triggering showed a marked increase in BM of Lineage⁻Sca1⁺cKit⁺ (LSK) cells (Fig. 1c). Among LSK cells, the percentage of MPPs was considerably increased in the BM during disease, together with the upstream and primitive CD34⁻CD48⁻CD150⁺ LT-HSCs (Fig. 1c). Of note, percentage changes in HSPC were mirrored by similar changes in absolute cell numbers (Supplementary Fig. 1e) because the total BM cellularity was unchanged during SpA (Fig. 1d).

We then investigated if the normal balance of differentiation among hematopoietic lineages was altered by chronic inflammation. Downstream of HSCs and MPPs, the BM of spondyloarthritic mice displayed a ~2.3-fold increase in the percentage and absolute number of granulocyte–monocyte progenitors (GMP) compared to healthy mice (Fig. 1e and Supplementary Fig. 1e). These GMPs are highly proliferative cells that give rise to neutrophils and monocytes/macrophages[20]. Bias toward myelopoiesis was confirmed functionally by an increase in the number of colony forming unit-granulocyte macrophage (CFU-GM) derived from BM in inflamed mice (Fig. 1e). The bias was also reflected in a decrease in megakaryocyte-erythroid progenitors (MEP), resulting in an increase in the balance of myeloid to erythroid progenitors in SpA (GMP:MEP ratio) (Fig. 1e). Greater myelopoiesis occurred at the expense of erythropoiesis and lymphopoiesis, with a significant decrease in common lymphoid progenitors (CLP) during disease (Fig. 1e). These changes in HSPCs correlated with a significant increase in mature BM neutrophils during SpA and a decline in BM erythroid and B cells compared to healthy mice (Fig. 1f). Ultimately, the increased BM myelopoiesis supported the chronic accumulation of short-lived neutrophils occurring in the paw joints and small intestine (SI) during SpA, i.e., ~50 and ~20-fold increases, respectively (Fig. 1g). These neutrophils had an activated phenotype in SpA, characterized by increased FcγRI expression, and production of TNF and IL-1β (Supplementary Fig. 1f), which are proarthritic cytokines[17].

In sum, the non-resolving joint and intestinal inflammation in spondyloarthritic mice was accompanied in the BM by dysregulated HSPC activity and biased differentiation toward myeloid progenitors, culminating in an invasion of target organs by activated and tissue-toxic neutrophils.

## Results

**Hematopoiesis is biased toward myelopoiesis during SpA.** Given that HSCs express receptors for inflammatory stimuli[7], we

**HSC and MPP upregulate myelopoiesis associated genes in SpA.** To identify potential modulators of HSPC activity during

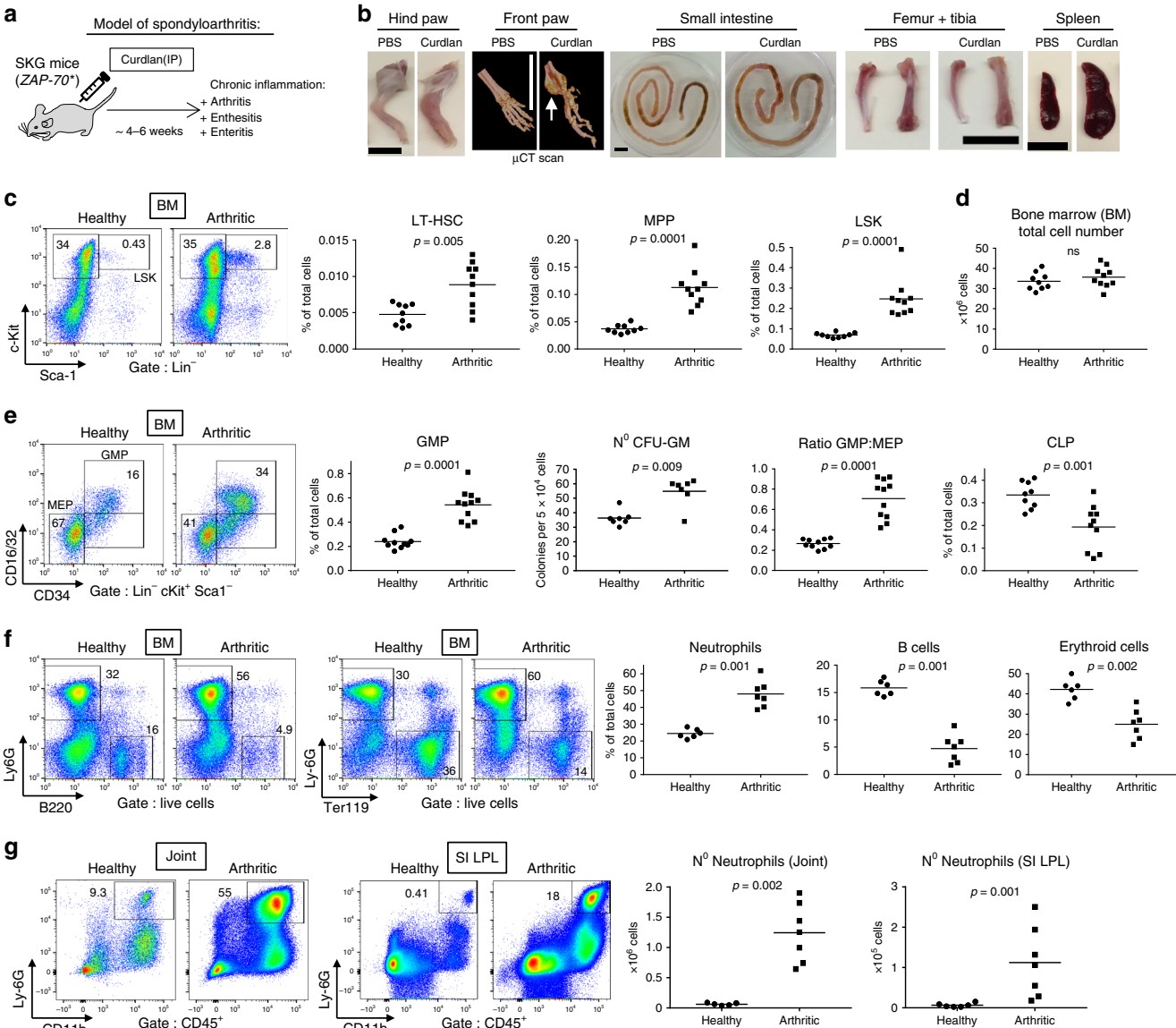

**Fig. 1 Hematopoiesis is biased toward myelopoiesis during experimental SpA. a** Experimental protocol to induce spondyloarthritis (SpA) in SKG mice. Single injection of curdlan IP causes non-resolving inflammation of joints, entheses, and small intestine (SI). Samples derived from such "arthritic" SKG mice culled 4–6 weeks after triggering were compared with PBS-injected "healthy" SKG mice (**b–g**). **b** Images of gross pathologic changes observed during SpA in SKG mice. Front paw: 3D-reconstructed ex vivo μCT radiographs of front paw. Arrow shows new bone formation characteristic of SpA. Scale bars = 1 cm. **c** Staining of bone marrow (BM) cells with frequencies of progenitors (Sca-1$^-$cKit$^+$) and LSK cells (Sca-1$^+$cKit$^+$) among Lin$^-$ cells. Graphs show frequencies of long-term hematopoietic stem cells (LT-HSC), multi-potent progenitors (MPP), and LSK cells among total cells. **d** Total count of BM extracted from one tibia and one femur of each mouse. **e** BM staining for CD16/32 and CD34, showing frequencies of granulocyte macrophage progenitors (GMP) and megakaryocyte-erythroid progenitors (MEP) among Lin$^-$cKit$^+$Sca-1$^-$ progenitor cells. Graphs show frequencies of GMP and common lymphoid progenitors (CLP), ratio of GMP:MEP, and number of myeloid CFU-GM colonies obtained from BM cells plated in methylcellulose medium. **f** Staining and graphs showing frequencies of mature neutrophils (Ly6G$^+$), B cells (B220$^+$), and erythroid cells (Ter119$^+$ red blood cells) among total BM cells. **g** Graphs and staining of cells from paws and small intestine (SI LPL), showing frequency and absolute number of neutrophils (CD11b$^+$Ly6G$^+$). Dots represent individual mice; horizontal bars indicate mean. Data are representative of three independent experiments (**b–g**). Groups were compared using Mann–Whitney U tests. Source data are provided as Source Data file.

inflammation, we studied the transcriptomes of FACS-sorted LT-HSCs, ST-HSCs, MPPs, and GMPs from the BM of spondyloarthritic and healthy SKG mice. As expected, analysis of RNA-sequencing data from each of these populations revealed a principal component of variation that captured the progression from multi-potency to lineage restriction (PC1, Supplementary Fig. 2a). Closer inspection of this component revealed that MPPs extracted from arthritic mice had a transcriptional profile more similar to GMPs than that displayed by their healthy counterparts. To investigate

this observation, we compared genes significantly differentially expressed between arthritic and healthy MPPs with those that showed significant differences between healthy MPPs and GMPs. We found a strong positive correlation (Spearman's rho = 0.75, $p = 2.2 \times 10^{-16}$) between transcriptional changes associated with SpA development in MPPs and the transcriptional differences between MPPs and GMPs in healthy SKG animals (Fig. 2a). This result suggests that MPPs are biased toward expression of pro-myeloid genes in SpA. For example, we found genes expressed at

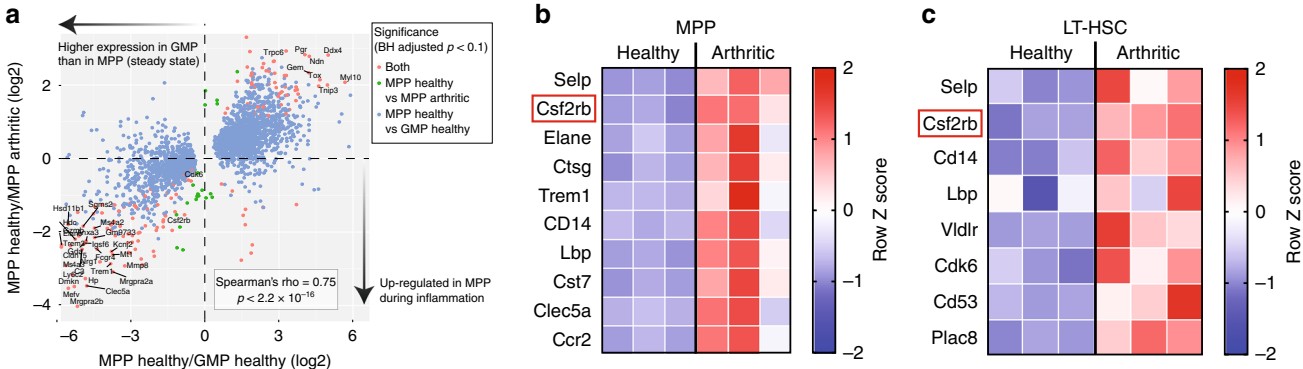

**Fig. 2 HSC and MPP upregulate myelopoiesis associated genes in SpA. a–c** In three separate experiments, mice ($n = 8$ per group) were injected with curdlan (arthritic) or PBS (healthy) before culling after 5 weeks and LT-HSCs, ST-HSCs, MPPs, and GMPs were isolated by FACS for RNA sequencing. **a** Scatter plot comparing genes differentially expressed between GMPs and MPPs in healthy animals (*x*-axis) with those differentially expressed in MPPs between healthy and spondyloarthritic animals (*y*-axis). **b, c** Heat map showing expression of selected genes upregulated in MPPs (**b**) and LT-HSCs (**c**) with development of SpA. Each column represents independent groups of eight pooled mice.

high levels by developing or mature neutrophils, e.g., *Elane* (elastase gene) or *Ctsg* (cathepsin G gene), and more broadly by myeloid cells, e.g., *Lbp, Cd14,* and *Trem1,* to be upregulated in the multi-potent MPPs in SpA (Fig. 2b and Supplementary Fig. 2b). In fact, some myeloid genes were already upregulated at the earliest stage of hematopoiesis, in LT-HSCs, e.g., *Lbp, Cd14,* and *Vldlr* (Fig. 2c and Supplementary Fig. 2b). Furthermore, we noted that *Cdk6,* which is essential for the egress of HSCs from quiescence[21], and *Plac8* and *Cd53,* which are involved in enhanced cell cycling[22,23], were upregulated on LT-HSCs in SpA (Fig. 2c), corresponding with the HSC increase observed in response to arthritis-derived inflammatory signals. Interestingly, the GM-CSF-receptor coding gene, *Csf2rb,* which is a pro-myeloid gene highly expressed by GMPs and mature myeloid cells[24], was upregulated on MPPs and LT-HSCs in SpA (Fig. 2b, c); this was confirmed at protein level (Supplementary Fig. 2c). Although the *Csf2ra* gene coding for the second subunit of the GM-CSF-receptor was expressed by LT-HSC, ST-HSC, and MPP, its levels were not increased during disease (Supplementary Fig. 2d).

As the precise stage at which GM-CSF acts on the hematopoietic tree (Supplementary Fig. 1d) is unclear and as we had detected expression of *Csf2ra and Csf2rb* in GMPs and also in MPPs and HSCs, we tested their responsiveness to GM-CSF. The myeloid cell output from GMPs cultured in pan-myeloid medium was substantially increased in response to GM-CSF (Supplementary Fig. 2e), and, interestingly, also from ST-HSCs and MPPs (Supplementary Fig. 2e and Supplementary Fig. 2f). This early stage responsiveness of HSPCs to GM-CSF was confirmed in vivo: treatment of curdlan-triggered SKG mice with GM-CSF for 7 days resulted in LT-HSCs, ST-HSCs, and MPPs increases (Supplementary Fig. 2g). Furthermore, GM-CSF treatment recapitulated the myeloid-skewed output from HSPCs observed with SpA, with a BM increase in GMP and neutrophils but decrease in MEP and mature erythroid cells and B cells (Supplementary Fig. 2g).

Together, these results revealed that the gene expression program of HSCs and MPPs is biased toward myeloid cell differentiation during SpA and that HSPCs are responsive to the pro-myelopoietic effect of GM-CSF as early as at the HSC and MPP stages, before the emergence of GMPs and mature myeloid cells.

## GM-CSF drives BM myelopoiesis and organ inflammation in SpA. After showing that HSCs and MPPs are responsive to GM-CSF, we examined the potential role of GM-CSF on

hematopoietic changes and clinical disease during SpA. CD4 T cells are an important source of GM-CSF[2,25]. Here, in SpA, we detected GM-CSF[+] CD4 T cells in joint-draining lymph nodes, but these cells were more abundant in the inflamed joints themselves (Fig. 3a). In addition, the percentage of GM-CSF producers among CD4 T cells and their absolute number inside the paw joints were significantly higher in spondyloarthritic compared to healthy SKG mice (Fig. 3a). GM-CSF[+] CD4 T cells were also substantially increased among small intestinal lamina propria leukocytes (SI LPL) during SpA, accompanied by increased intestinal levels of secreted GM-CSF (Fig. 3b).

Functionally, following triggering of SpA, prophylactic treatment of SKG mice for ~7 weeks with anti-GM-CSF antibody induced a striking decrease in LT-HSCs and MPPs in the BM compared to treatment with isotype control (Fig. 3c), confirming the GM-CSF responsiveness of HSPCs at early stages. Examining the effect of GM-CSF blockade on progenitors downstream of HSCs and MPPs, we observed decreased GMPs but increased CLPs, and a less substantial increase in GMP:MEP ratio associated with arthritis (Fig. 3d). Although total BM cellularity was not affected by anti-GM-CSF treatment (Supplementary Fig. 3a), HSPC changes induced by GM-CSF inhibition were accompanied by a significant reduction in neutrophils and increase of erythroid cells and B cells (Fig. 3e).

At peripheral sites, the myelopoiesis inhibition induced by GM-CSF blockade was followed by a substantial decrease in intestinal and articular neutrophil invasion (Fig. 3f). Strikingly, these changes were accompanied by significant ameliorations in the histological and clinical scores of arthritis (Fig. 3g), marked decreases in enthesitis-associated ankle thickening (Fig. 3g) and new bone formation (Supplementary Fig. 3b), and inhibition of the bone surface:volume ratio increase (Fig. 3g), highlighting an anti-GM-CSF-mediated reduction in inflammation-associated bone damage[26]. Similarly, in the intestine, mice treated with anti-GM-CSF exhibited a significant amelioration in enteritis and weight loss compared to isotype control treated mice (Fig. 3h and Supplementary Fig. 3c), highlighting the pathogenic role of GM-CSF in the development of the various symptoms of SpA. Of note, although the receptor subunit encoded by *Csf2rb* is common to receptors for GM-CSF, IL-3, and IL-5, treatment with anti-IL-3 or anti-IL-5 blocking antibodies did not inhibit disease (Supplementary Fig. 3d).

Considering potential clinical applications, we tested whether starting GM-CSF treatment only after the apparition of the first clinical sign of arthritis would prevent disease progression. Mice

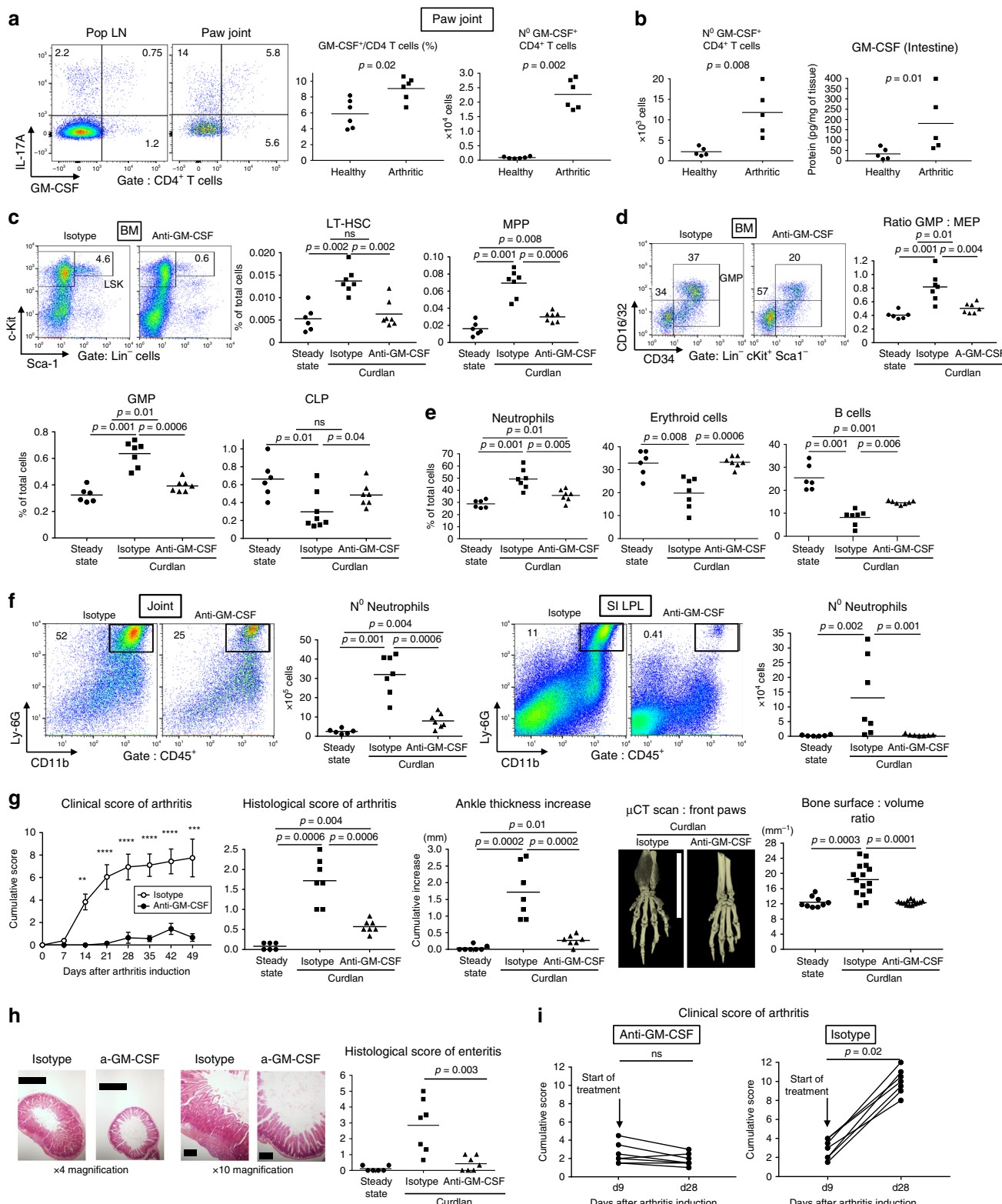

treated with an isotype control progressed to develop severe arthritis, while therapeutic injections of anti-GM-CSF starting from day 9 during ~3 weeks either stabilized or reduced the clinical score of arthritis (Fig. 3i).

**EMH occurs inside inflamed joints**. One obvious extra-articular manifestation associated with systemic inflammation in SpA was

splenomegaly (Fig. 1b), presaging the possible development of extramedullary hematopoiesis (EMH) during disease. In health, the majority of HSPCs reside in the BM but some recirculate in the blood and can accumulate in peripheral tissues during infection, producing inflammatory myeloid cells locally to contribute to immune defense[27,28]. In SKG mice, splenocyte numbers were ~twofold increased during disease (Fig. 4a). Although the percentage of GMPs among splenocytes was very low at steady

**Fig. 3 GM-CSF drives BM myelopoiesis and organ inflammation in SpA. a** Representative intracellular cytokine staining from CD4 T cells from popliteal lymph nodes and paw joints of mice culled 5 weeks after SpA induction. Frequencies and absolute numbers of articular GM-CSF⁺ CD4 T cells in healthy versus spondyloarthritic SKG mice; cells were incubated with brefeldin A, PMA and ionomycin before staining. **b** Absolute numbers of GM-CSF⁺ CD4 T cells in SI LPL and total GM-CSF protein in SI quantified by ELISA, n = 5 per group. **c–h** Curdlan-triggered SKG mice were treated for the entire duration of the experiment (6–7 weeks) with anti-GM-CSF or isotype control antibodies and compared to healthy (steady state) SKG controls. **c** LSK staining and frequencies of LT-HSCs and MPPs populations in BM. **d** GMP and MEP staining, GMP:MEP ratio, and frequencies of GMP and CLP populations in BM. **e** Frequencies of neutrophils, erythroid cells and B cells in BM. **f** Staining and absolute numbers of neutrophils in joints (left) and SI (right). **g** Mean arthritis scores (n = 7 mice in each group), histological arthritis scores and increase in ankle size (left). Representative μCT scans of isotype or anti-GM-CSF antibody treated SpA mice and changes in bone surface to volume ratio pooled from two independent experiments (right). Scale bars = 1 cm. **h** Representative photomicrographs of small intestinal sections (magnification ×4 and ×10) and histological enteritis scores. Scale bars = 1 cm (left panels) and 200 μm (right panels). **i** Clinical arthritis score of mice that received isotype or anti-GM-CSF antibody injections starting 9 days after injection of curdlan, when the first clinical signs of arthritis appeared (n = 7 mice in each group). Groups were compared using the Wilcoxon matched-pairs signed rank test. Data are representative of two independent experiments (**a–h**). Dots represent individual mice; horizontal bars indicate mean. Groups were compared using Mann–Whitney U tests unless noted otherwise. Source data are provided as Source Data file.

state, it was increased ~fourfold in SpA (Fig. 4a). Using a functional assay, we found that this increase in GMPs was mirrored by significantly higher numbers of CFU-GM derived from splenocytes of arthritic compared to healthy SKG mice (Fig. 4a), and a marked enrichment in splenic neutrophils (Fig. 4b).

As the inflamed joints in SKG mice are severely infiltrated by neutrophils and as EMH was occurring in spleen, we postulated that EMH might also develop directly inside the inflamed joints. Lin⁻cKit⁺ progenitors were, as expected, barely detectable in the soft tissue of uninflamed paws, but analysis of cells isolated from arthritic paw joints revealed the emergence of a GMP population with inflammation (Fig. 4c). This previously unrecognized phenomenon in chronic arthritis was substantiated by the high proliferation rate of GMPs accumulating in the inflamed joints, two thirds of articular GMP being Ki67⁺ (Fig. 4c).

To confirm that articular GMPs had the potential to contribute locally to myeloid cell accumulation, we next sought to assess their functional capacity. GMPs isolated from arthritic joints and cultured in a medium favoring myeloid cell differentiation generated ~70% CD11b^high myeloid cells in 4 days, which was similar to GMPs isolated from the BM, albeit with a lower total cell output (Fig. 4d). On average, ~60% of myeloid cells derived in vitro from articular or BM GMPs were Ly6G⁺ neutrophils (Fig. 4d), the remainder being CD11b⁺CD16/32⁺Ly6G⁻ monocytes/macrophages (Fig. 4d), while, as expected BM MEP or articular Lin^negcKit^neg cells were not able to generate myeloid cells (Supplementary Fig. 4a). Furthermore, GMPs isolated from the joints gave rise to CFU-GM similarly to BM GMPs, whereas Lin^negcKit^neg and mature neutrophils from the joints were incapable of such clonogenic activity (Fig. 4e). Commensurate with these in vitro observations, adoptive transfer of GMPs isolated from the BM of CD45.1⁺ mice into SKG mice (CD45.2⁺) confirmed that BM GMPs could migrate to inflamed joints and expand locally to give rise to tissue-toxic neutrophils, as also observed in the spleen (Supplementary Fig. 4b).

Since GM-CSF promoted experimental SpA, we next sought to investigate the role of GM-CSF in EMH development during disease. GM-CSF blockade significantly decreased percentages of GMPs and neutrophils in the spleen compared with isotype control treatment (Fig. 4f). Likewise, SKG mice treated with anti-GM-CSF displayed a significant reduction in articular GMP numbers and, functionally, a substantial decrease in CFU-GM activity in the joints (Fig. 4g).

In sum, these results reveal a previously unappreciated emergence of a highly proliferative and functional GMP population directly inside the joints during chronic inflammatory arthritis. GM-CSF was essential in promoting this articular EMH activity, which fueled the generation of tissue-toxic neutrophils directly inside the joints.

**MC contribute to GM-CSF production in SpA**. We next aimed to identify the main cellular sources of GM-CSF during disease. The intestinal lamina propria, which is inflamed in a high proportion of AS patients[11] and in curdlan-triggered SKG mice, is a major site of IL-23 production and a privileged location for Th17 cell and innate lymphoid cell (ILC) responses, which produce GM-CSF during inflammation[3,29–31]. Importantly, cytokines produced in the inflamed gut can reach systemic levels and remotely influence BM hematopoiesis, as described during intestinal infection[32,33].

With this in mind, we examined by FACS the potential cellular sources of GM-CSF in the intestine during SpA. Regarding the relative contribution of innate versus adaptive immune cells in inflammation, CD4 T-cell depletion revealed that these cells were not absolutely required for the development of experimental SpA (Supplementary Fig. 5a), fitting with the importance of innate cells described in AS[12,34]. In addition, as ILC and myeloid cells participate in the development of arthritic inflammation in SKG mice[35,36], we suspected a key role for innate cells in GM-CSF production. Although lymphoid cells are recognized as the main producers of GM-CSF[25], our analysis revealed that some intestinal Lin⁻Thy1⁻ cells, i.e., non-T cells and non-ILC, were also positive for GM-CSF (Fig. 5a). We hypothesized that MC, which are abundant at mucosal sites[37,38], were a possible candidate for this undefined source of GM-CSF. Indeed, the use of classical lineage markers does not identify MCs as they do not express CD11b or Gr1. Fittingly, lin⁻ckit⁺FcERI⁺ MCs from the intestine of inflamed mice expressed Csf2 (encoding GM-CSF), even to a greater extent than CD4 T cells (Fig. 5b); this was also observed in the joint (Fig. 5b). Using BM-derived MC (BMMC) (Supplementary Fig. 5b), which secrete high levels of inflammatory mediators[39], including IL-6 after LPS stimulation, we demonstrated in vitro that MCs could secrete GM-CSF upon similar activation (Fig. 5c).

Next, we confirmed that a high proportion of intestinal ckit⁺FcERI⁺ MC produced GM-CSF, to a similar extent as the Thy1⁺IL7R⁺ ILC (Fig. 5d) that were described recently as a major source of GM-CSF in our study of colitis[31] and in the joints in a model of rheumatoid arthritis (RA)[36]. Although arthritogenic TNF was produced by similar proportions of T cells, ILCs, and MCs, the percentage of GM-CSF⁺ cells was much greater among MCs and ILCs than CD4 T cells during SpA (Fig. 5d). In addition, in situ staining revealed that intestinal MCs were significantly enriched in the inflamed SI and joints in SpA (Fig. 5e and Supplementary Fig. 5d).

Therefore, besides ILCs and, to a lesser extent, CD4 T cells, our findings uncovered MCs as an unappreciated source of the pro-myeloid factor GM-CSF during SpA. We also showed that their

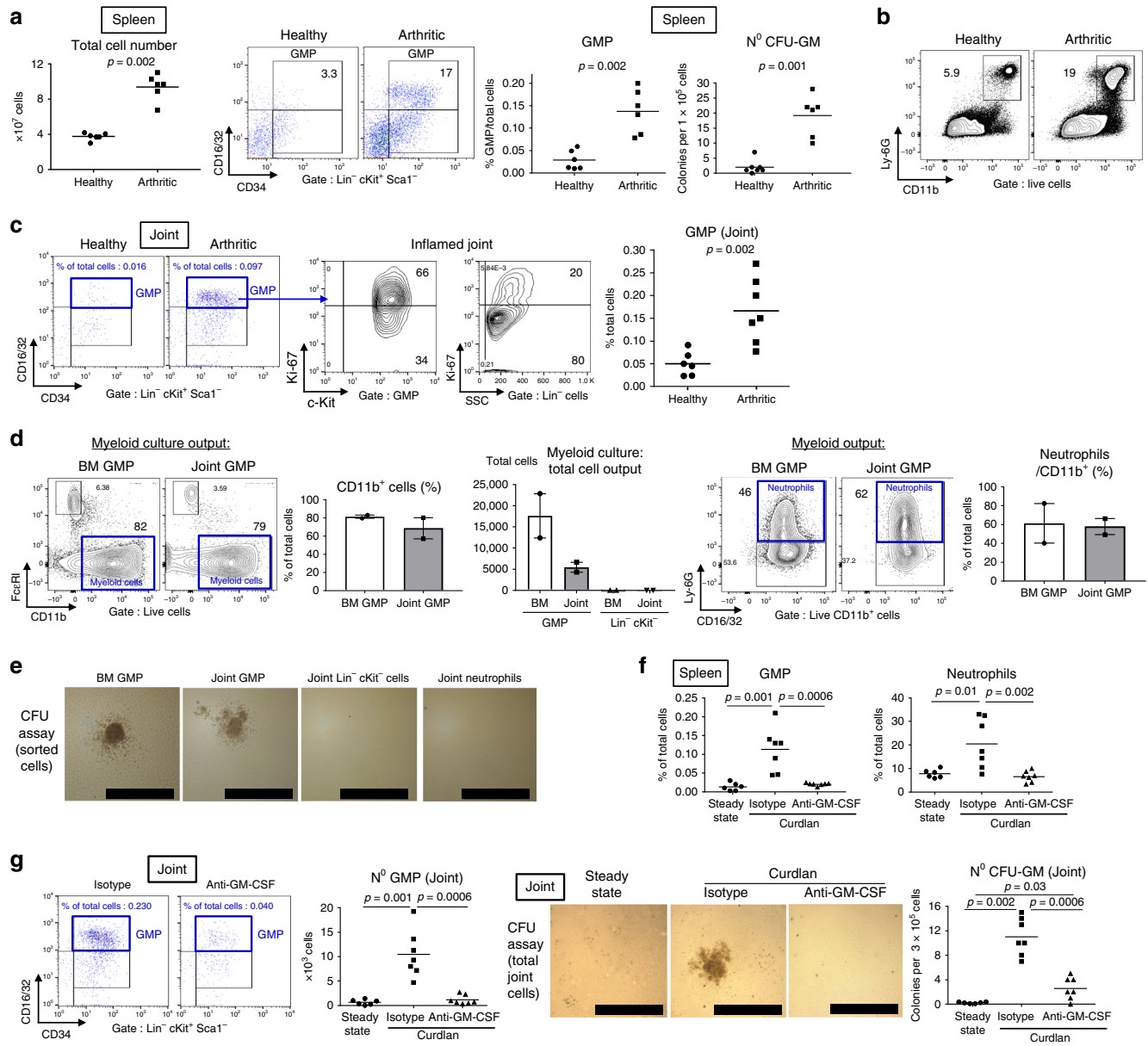

**Fig. 4 Extramedullary hematopoiesis occurs inside inflamed joints. a–c** Spondyloarthritic mice were culled (week 5) and compared with healthy SKG mice. **a** Total splenocyte numbers, GMP staining, frequencies of GMPs and CFU-GM numbers in spleen. **b** Representative neutrophil staining in spleen. **c** GMP staining, Ki-67 staining of GMP cells and control Lin⁻ cells, and frequencies of GMPs in paw joint single cell suspension. **d** Staining, frequencies of CD11b⁺ myeloid cells cultured from GMPs isolated from inflamed BM and joints and absolute numbers of myeloid cells following culture of indicated starting populations (left). Neutrophil staining following culture of GMPs from BM or joint and frequencies among myeloid cells (right). Bars represent mean and SEM of two independent experiments. **e** Representative photomicrographs of CFU-GMs formed from indicated cell populations. Scale bars = 500 μm. **f** GMP and neutrophil frequencies in spleens of healthy, isotype, or anti-GM-CSF antibody treated mice. **g** GMP staining and absolute numbers in joints at week 6–7 (left). Representative photomicrographs and numbers of CFU-GMs obtained from joint cells of indicated groups (right), scale bars = 500 μm. Data are representative of three (**a–c**) or two (**e–g**) independent experiments. Groups were compared using Mann–Whitney U tests. Source data are provided as Source Data file.

numbers were increased during disease, constituting a potential long-lasting source of local and systemic GM-CSF.

**MC are activated and their inhibition dampens SpA.** Previous studies of human inflammatory arthritis have described MCs as being activated in target organs and releasing inflammatory mediators such as IL-17A and TNF in AS and RA[40,41]. Because we found that MCs were increased in SpA and produced GM-CSF, we explored their contribution to disease development.

First, we confirmed our observation of increased MC numbers based on histological analysis of intestinal tissue by FACS, finding a significant increase in MC percentage among SI LPL of spondyloarthritic mice compared to uninflamed controls, and an increase of approximately fivefold in total numbers (Fig. 6a). Second, we found that intestinal MCs had an activated phenotype during disease, with increased surface expression of FcγRII/III and IL-33R compared to MCs from healthy mice (Fig. 6b). Of note, splenic MC numbers were also increased during SpA (Supplementary Fig. 5e).

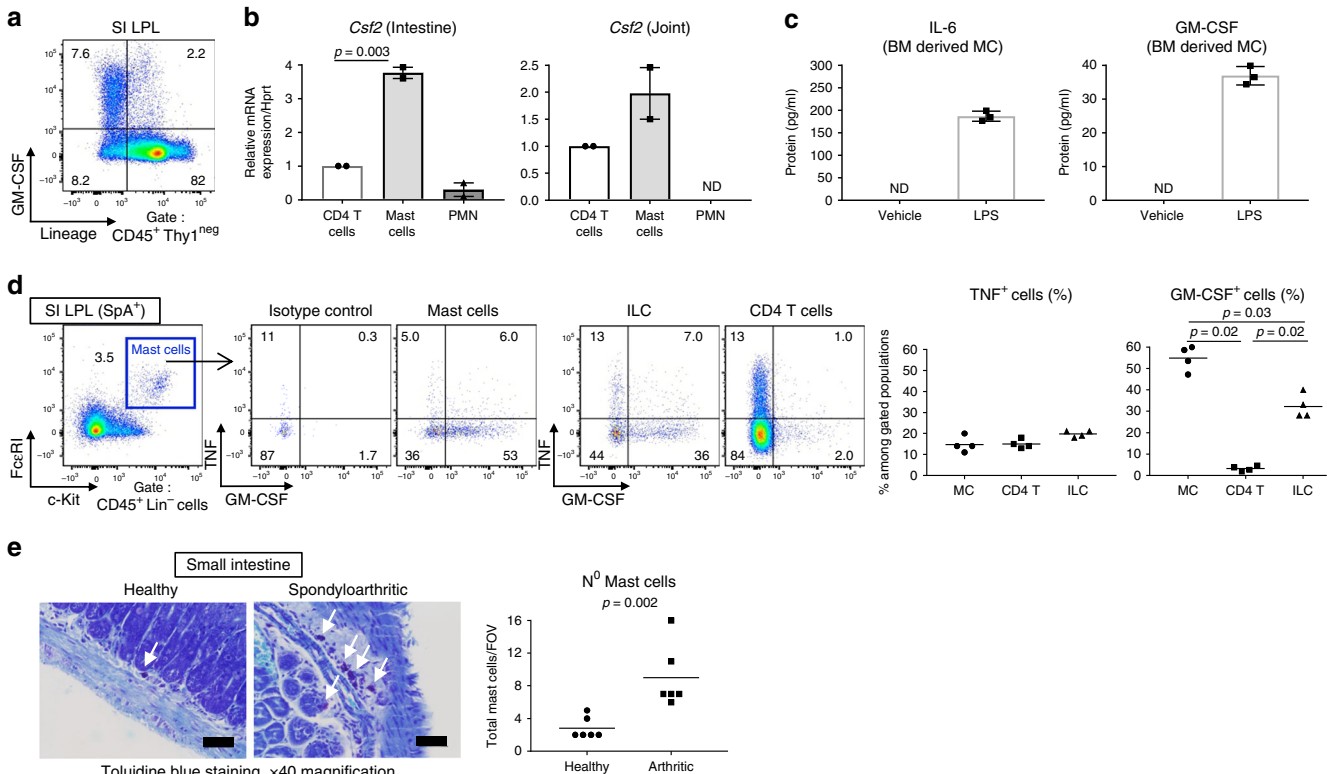

**Fig. 5 Mast cells contribute to GM-CSF production in SpA. a** Intracellular staining of GM-CSF+ cells in inflamed SI LPL from spondyloarthritic mice (week 4). **b** CD4 T cells, mast cells (MC) and neutrophils (PMN) were FACS-sorted from inflamed SI and joints and mRNA levels of *Csf2* were quantified by qPCR. Values are means relative to *Csf2* levels in CD4 T cells (set to 1). Bars represent mean and SEM of 2 separate experiments. ND not detected. **c** BMMCs were challenged with PBS or LPS for 16 h and cytokine levels were measured by ELISA. Bars represent mean and SD of technical triplicates. **d** MC staining in inflamed SI (week 4) and intracellular cytokine staining on MC, CD4 T cells, and ILCs (left). MC staining with an isotype control for GM-CSF is also depicted. Cells were incubated with brefeldin A only before staining. Frequencies of TNF+ and GM-CSF+ cells in indicated populations (right). **e** Photomicrographs of MCs in healthy and inflamed SI (week 4) by toluidine blue staining (scale bars = 50 μm) and absolute numbers of MCs per field of view (FOV). Data are representative of three (**c**, **d**) or two (**e**) independent experiments. Groups were compared using Mann–Whitney *U* tests. Source data are provided as Source Data file.

We next investigated the effect of MC inhibition on development of SpA. MC release various inflammatory compounds via two main distinctive pathways, i.e., degranulation of preformed compounds present in primary granules (e.g., histamine and prostaglandins) or release of de novo synthesized mediators (e.g., cytokines and chemokines)[42]. In our study, we used cromolyn, a well-described MC stabilizer, which inhibits MC degranulation and cytokine release[43,44]. Daily IP injection with cromolyn significantly dampened the SpA-associated increase in GM-CSF+ MC numbers, in the intestine and spleen (Fig. 6c), compared to PBS injection (Fig. 6d). Inhibition of MC activation was accompanied by decreased splenic and articular GMP numbers (Supplementary Fig. 5f) and also decreased neutrophil invasion of target organs (Fig. 6e). MC inhibition during development of SpA resulted in decreased paw and ankle swelling and diminished clinical score of arthritis (Fig. 6f).

In summary, inhibition of MC activity in SpA led to reduced numbers of GM-CSF-producing MCs and decreased joint inflammation, highlighting the contribution of MCs to the inflammatory network in SpA.

**IL-33 can promote GM-CSF production by MC during SpA.** We next resolved to identify potential triggers of MC activation and GM-CSF secretion in SpA. MCs can detect and respond to various inflammatory stimuli in peripheral tissues, including alarmins released upon tissue damage or mechanical stress, e.g.,

IL-33 and ATP[39]. IL-33 was of particular interest because its intestinal and joint levels were substantially increased in spondyloarthritic mice compared to uninflamed controls (Fig. 7a), as reported in the blood and intestine in AS patients compared to healthy controls[45,46]. Moreover, IL-33 promotes GM-CSF secretion by ILCs in mouse and human[36,47], and *St2*, which encodes IL-33R, was expressed at higher levels in MCs compared to other major immune cell types (Supplementary Fig. 6a). BMMCs likewise expressed high levels of IL-33R (Fig. S5c), as did MCs from intestinal tissues (Fig. 6b). In terms of MC responsiveness to IL-33, although it did not activate degranulation by BMMC as monitored by release of the granule component β-hexosaminidase (Supplementary Fig. 6b and ref. [48]), we demonstrated that IL-33 stimulation induced abundant GM-CSF secretion (Fig. 7b), in addition to its previously reported IL-6 promoting effect[39]. Increased GM-CSF levels were also detected intracellularly by FACS in activated BMMC following IL-33 exposure (Supplementary Fig. 6c). Of note, Cromolyn, which can inhibit cytokine secretion by MC[43,44], decreased IL-33 mediated GM-CSF release by BMMC (Supplementary Fig. 6d).

In light of these results, we hypothesized that IL-33 might promote dysregulated hematopoiesis and chronic inflammatory arthritis by stimulating GM-CSF secretion from activated MC. Importantly HSCs, MPPs, and GMPs do not express the IL-33R[49] and therefore cannot respond to any direct effect of IL-33. To evaluate IL-33 effects in SpA, we treated curdlan-triggered SKG males, which develop less severe disease than females[16], with

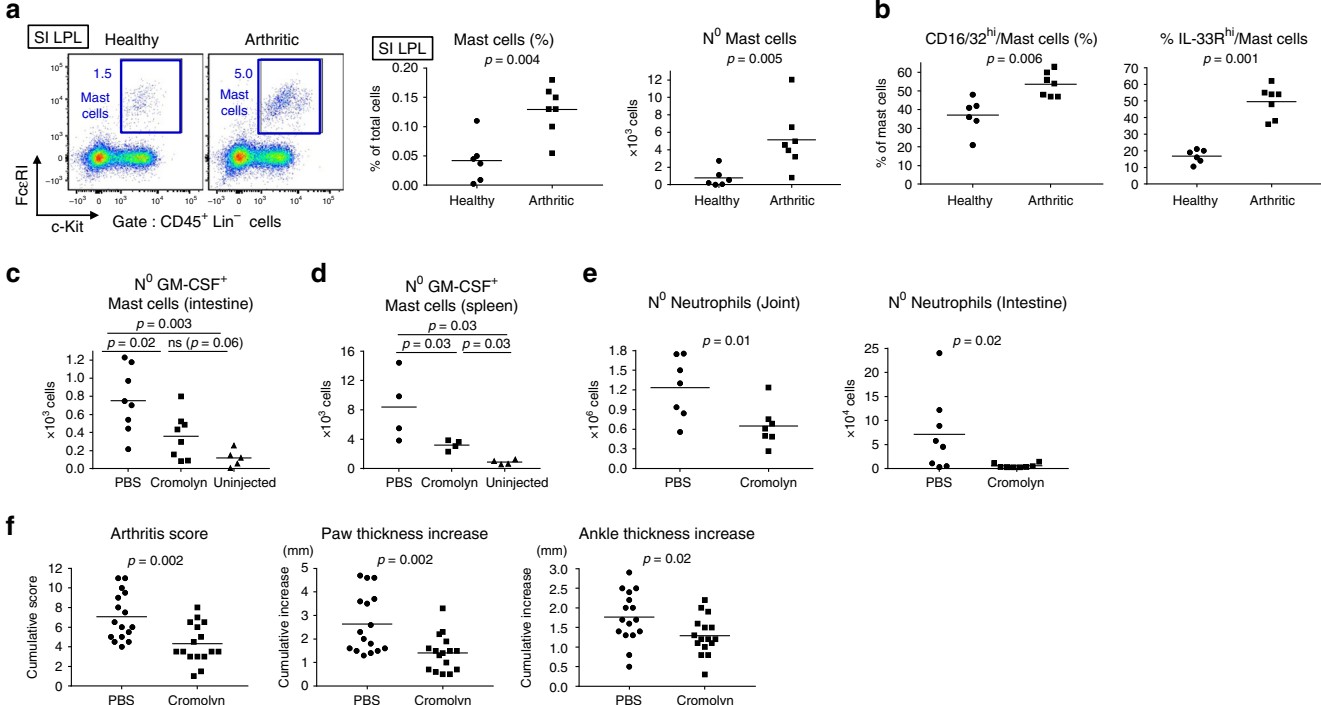

**Fig. 6 Mast cells are activated and their inhibition dampens SpA. a** FACS staining, frequencies and absolute numbers of MCs in the SI LPL of healthy vs. spondyloarthritic (week 5) SKG mice. **b** Percentage of CD16/32hi and ST2hi cells among MCs in SI LPL. **c–e** SKG mice were treated daily with PBS or cromolyn sodium following SpA induction and culled at week 4. Healthy SKG mice ("uninjected") were used as controls in (**c, d**). **c** Absolute numbers of GM-CSF+ MCs in the SI LPL. **d** Absolute numbers of GM-CSF+ MCs in the spleen. **e** Absolute numbers of neutrophils in joints and SI. **f** Clinical score of arthritis, ankle, and paw size increase. Dots represent individual mice; horizontal bars indicate mean. Data are representative of (**a–e**) or pooled from (**f**) two independent experiments. Groups were compared using Mann–Whitney U tests. Source data are provided as Source Data file.

twice weekly IL-33 injections for 5 weeks (Fig. 7c). This treatment led to increased levels of secreted GM-CSF in inflamed target organs (Fig. 7d). As reported before, IL-33 injections induced goblet cell hyperplasia and abundant intestinal mucus production[50], which precluded reliable isolation of viable SI LPL cells. Study of the MC response to IL-33 by FACS was possible in the spleen however, where MCs may also become activated and increased during inflammation, as reported in food allergy and sepsis[51,52]. With SpA, MCs were significantly expanded in the spleen of IL33-injected SKG mice compared to PBS-treated controls (Fig. 7e) and, remarkably, IL-33 treatment induced a substantial increase in GM-CSF+ cell percentage among MCs, from ~20% to ~60% (Fig. 7f). These IL-33-mediated changes in GM-CSF production were accompanied by a significant increase in BM LT-HSCs and MPPs (Fig. 7g) and biased differentiation toward myeloid cells. Indeed, a substantial increase in GMPs and neutrophil percentages occurred in the BM of IL33-treated mice compared to controls (Fig. 7g) even though the total BM cellularity was unchanged (Supplementary Fig. 6e). In addition, IL-33 injections promoted EMH because splenic and articular GMP numbers were substantially increased, with accompanying neutrophil expansion at both sites (Fig. 7h). Clinically IL-33 injections twice weekly with 1 μg of IL-33 induced a significant worsening in the clinical score of arthritis and the joint swelling severity (Fig. 7i), which was also observed with a twofold lower dose of IL-33 (Supplementary Fig. 6f).

Neutralization of IL-33 activity by ST2-Fc treatment did not affect levels of TNF produced by intestinal MC (Supplementary Fig. 6g), but resulted in a decrease in GM-CSF production (Fig. 7j), highlighting a promoting effect of endogenous IL-33 on GM-CSF secretion by MC. Although ST2-Fc treatment significantly decreased joint swelling and arthritis at early and

intermediate time points compared to mice injected with an isotype control (Fig. 7k and Supplementary Fig. 6h), the significance of this protective effect was lost at the final time point (Fig. 7k). This progressive loss of effect could be mediated, for example, by a compensatory effect of other cytokines able to promote GM-CSF secretion by MC, as described for TNF and IL-1β[53]. In line with the observed pathogenic role of GM-CSF in SpA, aggravation of arthritis and increased accumulation of GMP and neutrophils in inflamed joints induced by IL-33 injections were prevented if mice were co-injected with anti-GM-CSF (Fig. 7l and Supplementary Fig. 6i).

Collectively, our findings show that IL-33, of which endogenous levels were increased in SKG mice during SpA, has the capacity to promote the secretion of GM-CSF by MC, overall resulting in exacerbated HSC activity, biased differentiation toward GMP during SpA, and increased neutrophil invasion and inflammation of the joints.

## Discussion
HSCs have recently emerged as an integral part of the immune response and as central sensors of inflammatory stimuli during infectious and chronic inflammatory diseases[9]. Furthermore, prolonged exposure to inflammatory stimuli during chronic diseases may have long-lasting effects on the nature of the BM cell output through epigenetic modifications in HSPCs[54,55].

Here, we investigated the regulation of HSPCs in the context of chronic inflammatory arthritis to gain a greater understanding of the complex immune network in SpA. We observed that the differentiation of HSCs was markedly skewed toward GMPs, resulting in increased neutrophil production, which was mirrored by decreased lymphopoiesis and erythropoiesis. RNAseq analysis

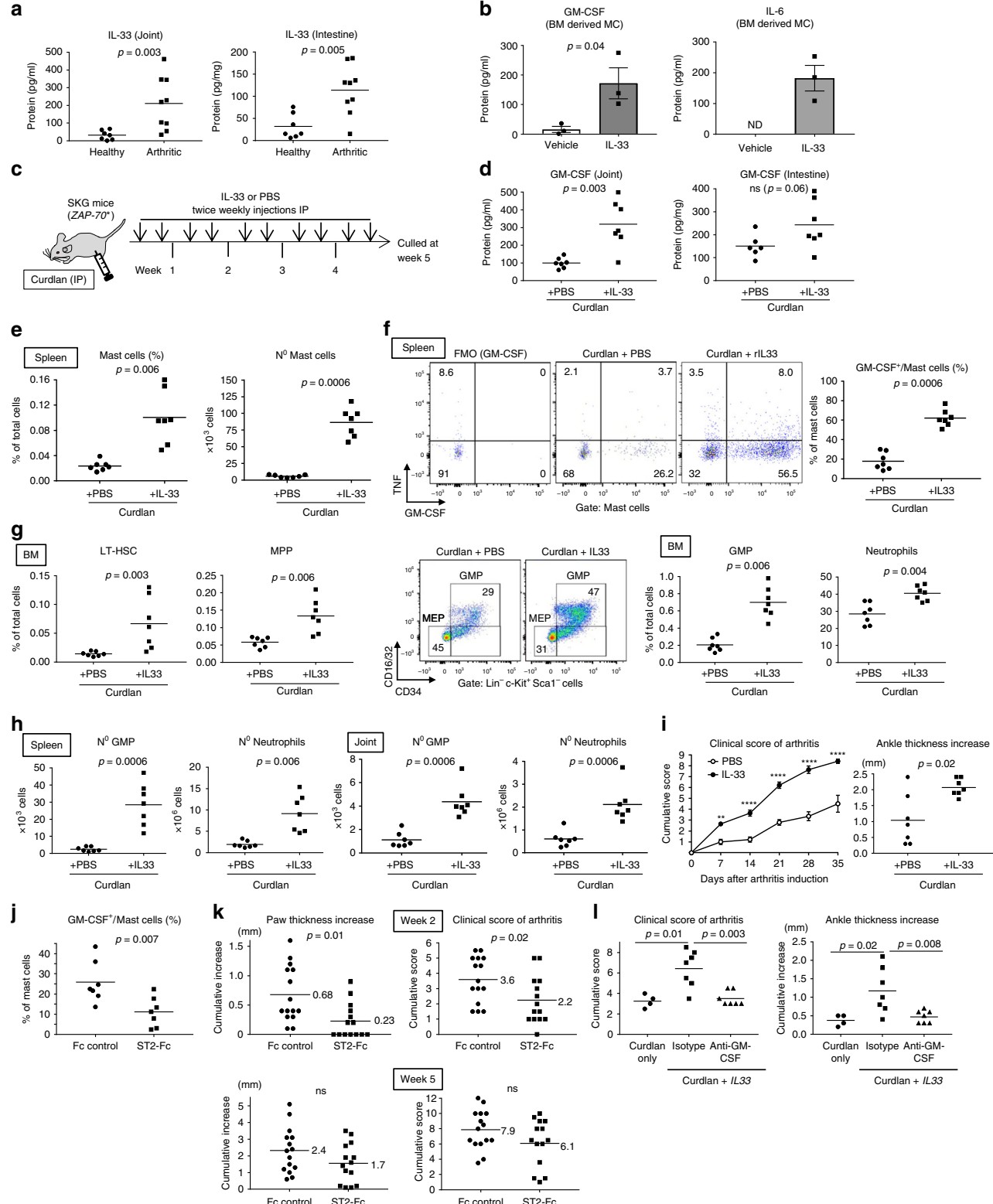

of HSCs and MPPs revealed a bias toward myeloid gene expression during SpA, e.g., *Csf2rb*. Early stage responsiveness of HSCs and MPPs to GM-CSF was demonstrated in vitro and in vivo. Its blockade in curdlan-triggered SKG mice significantly decreased myelopoiesis in the BM and at extramedullary sites, as well as intestinal and articular inflammation.

Beyond their paracrine and autocrine functions, cytokines can exert effects on distant organs, for example the BM, when they are produced at high levels in inflamed peripheral tissues during autoimmune or infectious diseases[15,32,56]. Approximately, 50–60% of AS patients develop subclinical intestinal inflammation[11], and the raised levels of various inflammatory cytokines observed in the intestine and blood of AS patients[29,45] could conceivably affect BM activity. Accordingly, while there have been no published studies describing hematopoiesis changes in the BM during AS, development of the

**Fig. 7 IL-33 can promote GM-CSF production by mast cells during SpA. a** IL-33 protein levels from joints and SI of healthy and spondyloarthritic mice, $n = 13$ mice per group. **b** BMMC were stimulated with PBS or IL-33 for 16 h and secreted IL-6 and GM-CSF quantified by ELISA. Error bars = SEM of 3 independent experiments. **c–i** SpA-triggered male SKG mice were injected IP twice weekly with PBS or 1 µg of rIL-33. **c** Experimental protocol. **d** GM-CSF protein levels from joints and SI from PBS or IL-33 treated mice, $n = 7$ mice per group. **e** Frequencies and absolute numbers of splenic MC. **f** Staining of intracellular cytokines (compared to GM-CSF FMO control) and frequencies of GM-CSF$^+$ cells among splenic MCs. **g** Frequencies of LT-HSCs and MPPs in BM (left). Staining of GMPs and MEPs in BM (middle). Frequencies of GMPs and neutrophils in BM (right). **h** Absolute numbers of GMP and neutrophils in spleens (left) and joints (right). Dots represent individual mice; horizontal bars indicate mean. **i** Mean arthritis scores ($n = 7$ mice in each group) and increase in ankle thickness. **j–k** SKG females were injected with ST2-Fc or control Fc fragment for 5 weeks after injection of curdlan. **j** Frequency of GM-CSF$^+$ mast cells in SI of Fc control- and ST2-Fc-treated mice after 5 weeks. **k** Change in paw size (left) and clinical arthritis score (right) in Fc control- and ST2-Fc-treated mice after 2 (top) or 5 (bottom) weeks. **l** SKG males were injected with curdlan then IL-33, with or without isotype or anti-GM-CSF antibodies. Dots represent individual mice; horizontal bars indicate mean. Clinical arthritis score (left) and change in ankle thickness (right) after 4 weeks. Data are representative of three independent experiments (**a–i**) or pooled from two independent experiments (**k**). Groups were compared using Mann–Whitney $U$ tests. Source data are provided as Source Data file.

disease is associated with increased blood neutrophil and decreased lymphocyte concentrations[57], such that the neutrophil: lymphocyte ratio has been used as an indicator of disease severity[58].

IL-33 is increased in the intestine and blood of AS patients compared to healthy controls[45,46], which echoes our observations in spondyloarthritic mice. IL-33R is expressed by MCs and type 2 ILCs[59], contrary to GMPs, MPPs and HSCs[49], rendering the latter cell types unresponsive to any direct effect of IL-33. Epithelial cells and fibroblasts release IL-33 in response to infection and tissue damage, which promotes cytokine secretion (e.g., IL-5, IL-9, and IL-13) by IL-33R-expressing leukocytes[59]. Besides these type 2 cytokines, our results indicate that IL-33 primes MCs to secrete GM-CSF, a cytokine often associated with type 17 immune responses[25] and, in our hands, acts directly on HSPCs to promote myelopoiesis in the SKG model of SpA, which is IL-17 and IL-23 dependent[60].

We previously observed dysregulated HSC activity, characterized by increased myeloid cell output, in murine models of colitis[2]. Similar phenomena were reported in mice with lupus[4] and atherosclerosis[61], revealing dysregulated HSC activity as a shared pathophysiological mechanism operating in diverse chronic inflammatory diseases. Interestingly, although BM samples from human patients suffering from chronic inflammatory diseases are scarce, people with atherosclerosis have increased CFU-GM activity in BM compared to matched controls[5], and RA patients have increased production of neutrophils but decreased production of erythroid cells in their BM[62,63].

GM-CSF was pathogenic in the SKG model of SpA and is emerging as a central player in chronic inflammatory diseases, including mouse models of multiple sclerosis[25], colitis[2] and RA[24,36]. In a zymosan-induced model of interstitial lung disease associated with arthritis development, GM-CSF blockade in SKG mice prevented and ameliorated lung and joint inflammation, while IL-17A blockade had only a minor effect on disease severity[64]. Interestingly, in our study GM-CSF not only drove dysregulated hematopoietic activity in the BM of spondylarthritic mice but also did so at extramedullary sites. We report the unexpected accumulation of highly proliferative and functional GMP directly in the inflamed joints of arthritic mice, which contributed to the accumulation of neutrophils in the joints during SpA. Although EMH is poorly understood, HSPCs were demonstrated to accumulate in various peripheral tissues in response to infection and generate inflammatory myeloid cells locally, e.g., granulocytes[27,28]. This phenomenon can contribute to fight infection[28], but is potentially detrimental to the host during non-resolving inflammatory diseases. For example, in atherosclerosis, splenic GMPs contribute to increased production of circulating monocytes and subsequent inflammation in atheromatous plaques[6].

ILCs and T cells are well-known sources of GM-CSF, as observed in AS[65]. Our data indicate that MCs also produce GM-CSF in experimental SpA, which builds on previous work showing that MCs are present in increased numbers in the synovial membrane of patients with SpA and RA, where they produce IL-17A and TNF[40,41]. In addition, MC were detected in the synovial fluid in a higher proportion of SpA patients compared to patients with other forms of arthritis[66]. Similarly to our observations in mice, a promoting effect of IL-33 on GM-CSF secretion by human MC has been reported in several studies[48,67], supporting the potential relevance of the MC-mediated IL-33/GM-CSF axis to the inflammatory response to humans. Consequently, MCs may represent a promising therapeutic target for SpA, with a recent pilot study suggesting this approach could be safe and effective[68]. In addition, our work highlights the potential benefits of antibody-mediated blockade of GM-CSF for SpA. Such treatment is safe in humans and has generated encouraging results in phase II clinical trials for RA[69].

Overall, our work sheds light on the complex inflammatory network in SpA, highlighting an underappreciated role for HSPCs in the co-ordination of inflammatory pathology at locations distant from the BM.

## Methods

**Mice and model of disease.** SKG mice (BALB/c-*zap70*$^{W163C}$), which are arthritis-prone because of a point mutation in ZAP-70[17], were bred and maintained under specific pathogen-free (SPF) conditions in licensed animal facilities at the University of Oxford. Mice were age matched across different treatment groups.

Where indicated, SpA was induced in 8–12-week-old SKG females by a single intraperitoneal (IP) injection of 0.3 mg of the microbial polysaccharide curdlan (Wako Chemicals USA, Inc.), which is a β-1,3-Glucan. Curdlan-injected spondyloarthritic mice ("arthritic") were compared to PBS-injected SKG mice ("healthy"), as unmanipulated or PBS-injected SKG mice do not develop arthritis under SPF conditions[16]. Although we did not investigate spinal inflammation in our study, this model also recapitulates the spondylitis phenotype characteristic of AS[16].

Disease was monitored and scored weekly. The scoring was done as follows: 0 = no swelling or redness, 0.5 = swelling or redness of digits and mild swelling and/or redness of wrists or ankle joints, 1 = mild swelling of wrist and ankle joints, 2 = moderate swelling of wrist and ankle joints, 3 = substantial swelling of wrist and ankle joints, not inhibiting normal mobility, feeding and drinking. Scores of forepaws and hind paws were totaled for each mouse. Paw thickness in the front paws, hind paws and ankles was also measured using dial-type callipers (Diatest Ltd., UK). Changes in paw or ankle size were calculated by subtracting the initial cumulated size of all paws or ankles from the cumulated size of all paws or ankles at the indicated time points.

All procedures involving mice were conducted according to the requirements of the UK Animals (Scientific Procedures) Act 1986. Experiments were approved by the University of Oxford Animal Welfare Ethical Review Body (Home Office Project License 30/3218).

**In vivo treatments.** Anti-GM-CSF (clone MP1-22E9) and isotype control antibody (clone 2A3) were purchased from Bio X Cell. Age matched female SKG mice were injected IP with either anti-GM-CSF antibody (0.25 mg) or isotype control antibody (0.25 mg) either on the day before curdlan injection and twice weekly for

the duration of the experiment (6–7 weeks) or 9 days after disease triggering and twice weekly until day 28.

Age matched SKG males were injected IP with recombinant murine IL-33 (1 µg; BioLegend) or PBS on the day before curdlan injection. IL-33 (0.5 and 1 µg) or PBS was administered twice weekly for the duration of the experiment (5 weeks). Notably, curdlan-triggered SKG males develop SpA more slowly and less severely than SKG females[16]. In a further experiment, male SKG mice were injected with curdlan then IL-33 (0.5 µg/injection) twice weekly, with anti-GM-CSF antibody (0.25 mg) or isotype control (0.25 mg) twice weekly for 4 weeks.

In a separate set of experiments, female SKG mice triggered for SpA development were injected IP daily either with the MC stabilizer cromolyn sodium (200 mg/kg; Sigma-Aldrich) dissolved in PBS or with PBS only for the duration of the experiment (28 days).

In a further experiment, female SKG mice were injected with curdlan then, after 1 week, with recombinant murine GM-CSF (2 µg; Peprotech) or PBS every 2 days for a total of 4 injections.

Female SKG mice were injected with curdlan, then ST2-Fc fragment (mouse ST2-mouse IgG1-Fc, Medimmune) or with a control mouse IgG1 (NIP228, Medimmune), both at 0.3 mg per IP injection three times weekly for 5 weeks.

Female SKG mice were injected with curdlan then anti-IL-5 antibody (Bio X Cell, clone TRFK5) or isotype (clone HPRN), both at 0.3 mg per injection IP three times weekly for 4 weeks. In a separate experiment, female SKG mice were injected with curdlan then anti-IL-3 antibody (Bio X Cell, clone MP2-8F8) or isotype (clone HPRN), both at 0.375 mg per injection IP twice weekly for 4 weeks.

Female SKG mice were injected with curdlan and anti-CD4 depleting antibody (Bio X Cell, clone YTS 191) or isotype (clone LTF-2), both at 100 mg per injection IP twice weekly for 4 weeks starting 2 days before curdan injection. The efficacy of the CD4 T-cell depletion was assessed using a different anti-CD4 clone (RM4-5).

**Histological scoring of arthritis and enteritis.** For histology, mice were culled and hind paws and sections of SI (duodenum, jejunum, and ileum) were fixed in 4% formalin before embedding in paraffin. The paws were decalcified with 10% neutral buffered EDTA for 4 weeks and then embedded in paraffin. Paraffin-embedded sections were cut (5 µm) and stained with hematoxylin and eosin; inflammation was scored in a blinded fashion. Histopathological severity of arthritis was scored in the tarsometatarsal, metatarsophalangeal, and inter-phalangeal joints by microscopy in a blinded fashion. The histological severity of arthritis was graded as follows: 0 = normal; 1 = minimal synovitis, cartilage loss, and bone erosions limited to discrete foci; 2 = synovitis and erosions present but normal joint architecture intact; and 3 = synovitis and extensive erosions present and joint architecture disrupted. The data are shown as the average score from the three joints for each mouse. Inflammation of the SI (epithelium, lamina propria, and submucosa) was scored in a blinded fashion using a previously described scoring system[2]. In indicated experiments, sections of SI and paws were stained with toluidine blue to visualize MC. MC were quantified by counting all cells in the section (SI) or by taking the average of cells counted in 8 × 20 objective fields in the periarticular regions of the tarsometatarsal and proximal interphalangeal joints of each mouse (paw).

**Cell isolation from SI for FACS analysis.** Mice were culled and SI were removed and washed in sterile PBS supplemented with 0.1% bovine serum albumin (BSA). SI was cut into sections and then opened longitudinally. Excess mucus was removed. Tissue was incubated in RPMI 1640 supplemented with 5% fetal calf serum (FCS) and 2 mM EDTA on a shaking incubator (150 rpm) heated to 37 °C for 20 min. This EDTA incubation was repeated for 20 min, after which, pieces of tissue were incubated in RPMI 1640 supplemented with 5% FCS at room temperature for 10 min without shaking. Pieces of tissue were then transferred to new tubes and digested in RPMI 1640 supplemented with 10% FCS, DNaseI (40 U/ml; Sigma-Aldrich) and collagenase VIII (0.25 mg/ml; Sigma-Aldrich) for 30 min at 37 °C on a shaking incubator. Tissue was filtered through a 70 µm strainer and centrifuged. Digested tissue was separated by centrifugation on a 30%/40%/70% Percoll gradient (GE Healthcare), centrifuging at 1800 rpm for 20 min with no brake. Cells at the 40%/70% interface were collected as the leukocyte-enriched fraction, i.e., SI LPL.

**Cell isolation from spleen and BM for FACS analysis.** Spleens, femurs and tibiae were harvested into PBS supplemented with 0.1% BSA. Spleens were passed through a 70 µm filter. The single cell suspension was centrifuged and resuspended in 1 ml red blood lysis solution for 2 min at room temperature. Lysis was stopped by addition of ice-cold PBS with 0.1% BSA. Cells were centrifuged and resuspended in PBS with 0.1% BSA before being stained for FACS analysis. BM cell suspensions were prepared by flushing the marrow from femur and tibia and were resuspended in PBS with 0.1% BSA. Cells were centrifuged and re-suspended, before staining for FACS analysis.

**Cell isolation from paw joint for FACS analysis.** The skin was removed from one hind paw and the paw was cut into four pieces. Pieces of tissue were incubated in 1 ml of RPMI 1640 supplemented with DNAse I (0.1 mg/ml) and Liberase (Roche, 0.3 mg/ml) for 90 min at 37 °C. Digested soft tissue from the paw joint was then passed through a 70 µm strainer to obtain a single cell suspension which was stained for flow cytometry.

**Flow cytometry and cell sorting.** Single-cell suspensions of desired tissues were obtained as described above. In total, $1-2 \times 10^6$ cells were stained with fixable viability dye (BioLegend) for 15 min in the dark at room temperature. Cells were centrifuged and incubated with unlabeled anti-CD16/32 to block nonspecific staining (unless anti-CD16/32 was in the staining panel), and were then stained with surface antibodies in PBS with 0.1% BSA for 30 min in the dark at 4 °C. Cells were centrifuged and fixed for 15 min using fixation solution (Cytofix/Cytoperm, BD) at 4 °C. Cells were centrifuged and re-suspended in PBS with 0.1% BSA and 2 mM EDTA.

For intracellular staining, cells were first incubated in RPMI 1640 with 10% FCS and 5 µg/ml brefeldin A with or without PMA (5 ng/ml) and ionomycin (500 ng/ml) for 3 h at 37 °C and 5% $CO_2$. Cells were stained with a viability dye and for surface antigens as above for 30 min. After fixation, cells were permeabilized by addition of cytoperm solution for 15 min followed by addition of intracellular antibodies diluted in cytoperm solution for 45 min at 4 °C in the dark. Cells were centrifuged and resuspended in PBS with 0.1% BSA and 2 mM EDTA before acquisition.

The following monoclonal antibodies were used for flow cytometry analysis, with the name of the clone indicated in brackets: anti-CD45 (clone 30-F11), anti-CD11b (M1/70), anti-Ter119 (Ter119), anti-Gr-1 (RB6-8C5), anti-CD3 (17A2), anti-CD4 (clone RM4-5 or clone GK1.5), anti-NKp46 (29A1.4), anti-CD11c (N418), anti-B220 (RA3-6B2), anti-FcεR1 (MAR-1), anti-Ly-6G (1A8), anti-Ly6C (HK1.4), anti-F4/80 (BM8), anti-CD16/32 (i.e., FcγRII/III; clone 2.4G2), anti-CD90.2 (30-H12), anti-CD117 (2B8), anti-CD34-FITC (RAM34), anti-Ly-6A/E (i.e., Sca-1; clone D7), anti-CD150 (TC15-12F12.2), anti-CD48 (HM48-1), anti-GM-CSF (MP1-22E9), and isotype control: Rat IgG2a (RTK2758), anti-TNF-α (MP6-XT22), anti-pro-IL-1β (NJTEN3), anti-IL-33Rα (i.e., ST2; clone DIH9), anti-Ki67 (16A8), anti-IL-7R (A7R34), anti-CD64 (i.e., FcγRI; clone X54-5/7.1). Antibodies were purchased from BioLegend, BD Biosciences or eBioscience. Viable cells were identified as unstained with Zombie Aqua or Green (BioLegend).

Samples were acquired on an LSR II or LSR-Fortessa (Becton Dickinson) and analyzed using Flowjo Software (TreeStar Inc). Cell sorting was performed using a FACS ARIA III (Becton Dickinson).

**Quantitative real-time PCR.** Total RNA was isolated from sorted cells using the RNeasy micro kit (QIAGEN) and 50 ng of total RNA was reverse transcribed using the High-Capacity cDNA Reverse Transcription Kit (Affymetrix eBioscience). Total RNA from SI was isolated by tissue disruption using lysis beads in RLT buffer and the Precellys24 homogenizer. RNA was isolated using the RNeasy Mini Kit (QIAGEN) according to manufacturers' instructions. Quantitative RT-qPCR was performed using Taqman assays and PrecisionFast Mastermix (PrimerDesign) on a ViiA7 384-well real-time PCR system. All expression levels were normalized to an internal reference gene (*Hprt*) and calculated as $2^{-(CT\ Hprt-CT\ gene)}$. Real-time qPCR was performed in duplicate and using the following Taqman primers (Thermo Fisher Scientific: Csf2 (Mm01290062_m1), Tnf (Mm00443258_m1), Hprt (Mm03024075_m1), Il6 (Mm00446190_m1), Il23 (Mm00518984_m1), Il1b (Mm00434228_m1), and St2 (Mm00516117_m1).

**RNA sequencing.** SpA was induced in 8–10-week-old female SKG mice by injecting curdlan IP and mice were culled 5 weeks after disease triggering. Healthy or spondyloarthritic SKG mice were culled and LT-HSCs, ST-HSCs, MPPs, and GMPs were FACS-sorted from BM directly into RLT buffer containing β-mercaptoethanol. LT-HSCs were defined as Lineage⁻, c-kit⁺, Sca-1⁺, CD34⁻, CD150⁺, CD48⁻, ST-HSCs were defined as Lineage⁻, c-kit⁺, Sca-1⁺, CD34⁺, CD150⁺, MPPs were defined as Lineage⁻, c-kit⁺, Sca-1⁺, CD34⁺, CD150⁻, CD48⁺, and GMPs were defined as Lineage⁻, c-kit⁺, Sca-1⁻, CD34⁺, CD16/32$^{hi}$. Of note, ST-HSC are also referred to in other publications as "MPP1-2" and MPP as "MPP3-4"[19]. RNA was isolated using the RNeasy Micro Kit (QIAGEN) according to manufacturers' instructions. cDNA was prepared using the SMARTer protocol (Clontech) and sequencing libraries prepared using the NEBNext Ultra protocol (NEB). Libraries were subject to 75 bp paired-end sequencing (Illumina HiSeq 4000) to an average depth of 11.2 million read pairs per sample.

**Computational analysis of RNA sequencing.** Sequence reads were aligned to the mouse genome with Hisat2 (version 2.0.4) using a "genome_trans" index built from the mm10 release of the mouse genome and Ensembl version 83 annotations (two-pass strategy to discover novel splice sites; with parameters: –dta and –score-min L,0.0,−0.2). Mapped reads were counted using featureCounts (Subread version 1.5.0; Ensembl version 83 annotations; default parameters). FPKMs were estimated with cuffquant (Cufflinks version 2.2.1; Ensembl version 83 annotations; with parameters: –multi-read-correct –no-effective-length-correction –max-bundle-frags 2000000 –max-mle-iterations 10000 –frag-bias-correct). Upper quartile normalized FPKM values were computed using cuffnorm (Cufflinks version 2.2.1; with parameters: –compatible-hits-norm library-norm-method quartile). Differential gene expression analysis was performed using DESeq2 (dispersions estimated with a "local" fit). Principal components analysis (PCA) was performed using

normalized and transformed counts (variance stabilizing transformation; computed with DESeq2) and the R "prcomp" package. The PCA was performed using only genes that showed significant variation between the experimental groups (i.e., all possible combinations of cell type and condition; DESeq2 LRT test; BH adjusted $p < 0.1$).

**MicroCT analysis**. Hind and front paws of indicated mice were removed and fixed in 4% formalin. Paws were scanned using a SKYSCAN 1174 Compact Micro-CT machine (Skyscan, Bruker, Belgium), 50 kV, 800 µA, 8.3 µm isometric voxel resolution, 0.7° rotation step with 0.5 mm Al filter and exposure set to 2000 ms and at 12.57 µm image pixel size. Images were analyzed using Skyscan CT Analyzer software version 1.13.2.1.

**BMMC culture**. Total BM cells were isolated by flushing the BM from femur and tibia of healthy SKG mice using RPMI 1640 supplemented with 10% FCS. A single cell suspension was obtained and passed through a 70 µm strainer before resuspending in complete growth medium (RPMI 1640 supplemented with 10% FCS, 1 mM sodium pyruvate, 4 mM L-glutamine, 25 mM HEPES, 50 µM 2-ME, 10 ng/ml murine IL-3, nonessential amino acids and 10,000 U/ml penicillin/streptomycin) and cells were seeded into tissue culture treated T75 flasks (Corning). Two days after seeding, non-adherent cells were harvested and transferred into fresh medium in a new tissue culture flask; this was repeated twice weekly. Cell density was adjusted to a concentration of $1 \times 10^6$ cells/ml after the first 2 weeks. Cells were cultured for a total of 5–6 weeks to obtain a pure population of MC. Cell purity of ~95% was obtained as assessed by co-expression of the markers c-Kit and FcεRI by flow cytometry.

**BMMC activation assays**. For cytokine production assays, BMMC were stimulated with vehicle, LPS (100 ng/ml; Sigma-Aldrich) or recombinant murine IL-33 (50 ng/ml; BioLegend) for 16 h, with or without cromolyn sodium (5 mM). Supernatant was harvested and indicated proteins were quantified by ELISA according to manufacturers' instructions. In some cases, brefeldin A (5 µg/ml) was added for a further 6 h before cells were stained for cytokine production and assessed by flow cytometry, as described above. β-hexosaminidase release was measured as described previously[70]. Briefly, $5 \times 10^4$ cells were re-suspended per well in a 96-well plate in HEPES buffer and rested for 10 min at 37 C before adding stimulants (IL-33 (50 ng/ml) or PMA (5 ng/ml) and ionomycin (500 ng/ml)) for 30 mins. After this, the plate was centrifuged at $300\,g$ for 5 mins and 50 µl of supernatant was transferred to a plate containing 100 µl p-nitrophenyl N-acetyl-β-D-glucosamide (PNAG, Sigma) solution (3.5 mg/ml) per well. In the original plate, 150 µl of 0.1% Triton X-100 solution (Sigma) was added to each well and mixed 5 times before incubating for 10 min at 37 °C. After this, 50 µl of the lysed cell solution was transferred to a different PNAG plate. Both PNAG plates were incubated at 37 °C for 90 min before adding 100 µl glycine solution (Sigma). Light emission was measured with a plate reader at 405 nm, subtracting emission at 620 nm. Release was expressed as a percentage of the total β-hexosaminidase present in each sample after lysis. All activation assays were completed in triplicate.

**Liquid culture of BM HSPCs**. Cells (LT-HSCs, ST-HSCs, MPPs, and GMPs) were sorted for culture from suspensions of cells derived from BM using the gating strategies outlined above, in "RNA sequencing". Cells were cultured in 96-well tissue culture plates, with 500 HSC and 1000 MPP or GMP added to each well in 200 µl of IMDM medium containing L-glutamine and HEPES and supplemented with 5% FCS, 2-mercaptoethanol (100 µM), penicillin/streptomycin (10,000 U/ml), nonessential amino acids (1×), sodium pyruvate (1 mM), murine stem cell factor (20 ng/ml), murine IL-6 (10 ng/ml), murine IL-11 (10 ng/ml), murine IL-3 (10 ng/ml), murine thrombopoietin (10 ng/ml), and human erythropoietin (10 ng/ml). All cultures were performed in triplicate, with some also containing recombinant murine GM-CSF (20 ng/ml). Cells were cultured for 4 days (GMP) or 6 days (LT-HSCs, ST-HSCs, and MPPs), with 60 µl of medium removed and replaced with fresh medium every 2 days. After 4–6 days, cells were prepared for flow cytometry as described above. The myeloid output of stem cells or progenitors was assessed by expressing the number of CD11b^hi cells as a percentage of the live cells harvested at the end of the culture. Cells were quantified using a hemocytometer. Of note, the majority of CD11b^hi cells were also CD16/32⁺.

**Myeloid output assays from GMPs sorted from the joints**. GMPs were sorted from digested paws by FACS, as described above, and compared to GMPs sorted from flushed BM. Articular lineage⁻c-Kit⁻ cells and MEP from BM were also obtained as controls. The digested tissue of 12 paws from mice injected with curdlan 4 weeks previously were pooled to obtain a suitable number of cells, and the BM derived from 4 femurs and 4 tibiae from the same mice were also pooled for isolation of BM GMP. These cells were cultured in 96 well tissue culture plates in IMDM medium containing L-glutamine and 25 mM HEPES and supplemented with 5% fetal bovine serum, MEM nonessential amino acids, penicillin/streptomycin (10,000 U/ml), sodium pyruvate (1 mM), β-mercaptoethanol (50 µM), recombinant murine GM-CSF (10 ng/ml), recombinant murine IL-9 (50 ng/ml), recombinant murine IL-3 (20 ng/ml), and recombinant murine stem cell factor (20 ng/ml). Cell suspensions were all in a total volume of 200 µl, with 1000 cells

added to each well at the beginning of the culture. All cultures were performed in triplicate. Suspensions were maintained at 37 °C in a 5% $CO_2$ atmosphere, and 60 µl of the medium was removed and replaced with fresh medium every 2 days. After 4 days, cells were quantified using a hemocytometer. The culture plates were also centrifuged and resuspended in fixable viability dye for 15 min at 4 °C. The cells were then washed by adding 100 µl PBS with 0.1% BSA and prepared for myeloid cell output analysis by flow cytometry, as described above.

**GMP transfer experiments**. Total BM cells were isolated from femurs and tibiae of CD45.1⁺ BALB/c mice (CByJ.SJL(B6)-*Ptprc*^a, the Jackson Laboratory) and GMP were isolated using FACS, as described above. In total, $5 \times 10^4$ GMP were transferred intravenously to congenic CD45.2⁺ SKG mice 2 weeks after IP injection of curdlan to trigger inflammation in the joints. CD45.1⁺ cell infiltration to the inflamed joints and spleens and differentiation into neutrophils were investigated using flow cytometry.

**CFU assays**. CFU-GM derived from FACS-sorted cells were assessed by mixing 1000 sorted GMP from paw or BM in a volume of 100 µl IMDM (with L-glutamine and 25 mM HEPES) with 1 ml of methylcellulose medium (MethoCult M3434 containing murine stem cell factor, IL-3, IL-6, and EPO). The suspension was mixed by pipetting and vortexing and applied to 3 cm petri dishes. These dishes were incubated at 37 °C and 5% $CO_2$. Images of the colonies were acquired after 6 days by bright field microscopy using an Olympus BX51 microscope. Colony formation assays were performed in duplicate.

For CFU-GM quantification from organ single cell suspensions, BM cells ($5 \times 10^4$ cells), splenocytes ($1 \times 10^5$ cells) or cells isolated by enzymatic digestion of the soft tissue of the paw joint ($3 \times 10^5$ cells) were suspended in methylcellulose medium (Stem Cell Technologies; M3434) to quantify clonogenic myeloid colony forming units. Morphologic analysis of colony formation was performed after 10–12 days of incubation.

**Ex vivo organ explant experiments**. Three sections of SI (duodenum, jejunum, and ileum) (0.25 cm²) and one forepaw were removed and incubated in RPMI 1640 supplemented with 10% FCS and 10,000 U/ml penicillin/streptomycin at 37 °C and 5% $CO_2$ for 16 h. The forepaw was cut into three pieces before incubation. Supernatant was harvested and indicated proteins were quantified by ELISA according to manufacturers' instructions. Intestinal cytokine concentrations were normalized to explant weight. Articular concentrations were expressed as total cytokine levels released per paw.

**Statistical analysis**. Statistical analysis was performed with Prism 7.0 (GraphPad Software). Unless stated otherwise, the nonparametric Mann–Whitney test was used for all statistical comparisons. Differences were considered statistically significant when $p < 0.05$.

**Reporting summary**. Further information on research design is available in the Nature Research Reporting Summary linked to this article.

## Data availability

All data associated with this study are available in the main text or the Supplementary materials. The source data underlying Figs. 1c–g, 3a–i, 4a, c, d, f, g, 5b–e, 6a–f, 7a, b, d–l, and Supplementary Figs. 1a, b, c, e, f, 2c–g, 3a, c, d, 5a, d–f, 6a–h are provided as a Source Data file. The RNA-sequencing data have been deposited in GEO (GSE126218).

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

## Acknowledgements

We thank the High-Throughput Genomics Group at the Wellcome Trust Centre for Human Genetics (funded by Wellcome Trust grant reference 090532/Z/09/Z) for the generation of the Sequencing data. We thank Shimon Sakaguchi (Osaka University) for providing SKG mice, Catherine Huntington (MedImmune), Jonathan Webber (University of Oxford) and Ida Parisi (University of Oxford) for technical assistance, and Roland Kolbeck (MedImmune), Marc Feldmann (University of Oxford) and Annette Plüddemann (University of Oxford) for helpful discussions. This study was funded by Versus Arthritis (Career Development Fellowship 20834 to T.G., and Senior Fellowship 20372 to N.J.H.) and the Kennedy Trust for Rheumatology Research.

## Author contributions

T.G., D.R.K., J.W.S., P.D., and S.N.S. performed and analyzed the experiments. T.G., D.R.K., J.W.S., and S.N.S. wrote the paper. E.S.C and N.J.H. provided essential materials and were involved in the data discussions. T.G. supervised, conceived, and designed the study.

## Competing interests

S.E.C. is employed by MedImmune/AstraZeneca. She provided ST2-Fc and control mouse IgG1 reagents, as well as scientific input in the design of the related experiment. The remaining authors declare no competing financial interests.
