## [Peer Review File · Nature Communications]

Reviewers' comments:

Reviewer #1, expert on IL-33 (Remarks to the Author):

This is a very well-presented study with convincing data addressing an important disease.

That Neutrophils driven by GM-CSF via myelopoiesis is important in arthritic diseases has been well-documented. GM-CSF is produced by several cell types, including Mast cells. The present study links IL-33 to MCs which produce GM-CSF that drives myelopoiesis that recruit neutrophils to the inflammatory joints. Infection is also an important factor. SKG mice do not develop inflammatory disease in a germ-free environment. Several questions emerge from this study.

1. How important is this axis compared to other pathways, such as CD4 cells in SpA? IL-33 was given in relatively high quantity to induce SpA (1 ug, 2x weekly for 5 weeks). Neutralisation of IL-33 or IL-33 or ST2 ko mice crossed to SKG mice might resolve this issue.
2. How do HSCs recruit neutrophils?
3. How relevant are the data to clinical disease in humans? The author cited studies reporting increase of IL-33 in SpA patients. In vitro studies showing that IL-33-MC-GM-CSF pathways works in human cells would also be helpful.

Reviewer #2, expert on expert on cytokines in autoimmunity (Remarks to the Author):

This is an interesting report particularly the detection of GMP in inflamed joints, the putative role of GM-CSF+ mast cells and the blockade of the arthritis with anti-GM-CSF mAb.

Inflammatory arthritis

Major Comments

In general, the paper would have been strengthened by showing whether therapeutic blockade with anti-GM-CSF mAb is effective in addition to the prophylactic protocol exclusively adopted – the former is more relevant to potential clinical applications in SpA which they suggest.

Another weakness is that IL-33 was only used exogenously to exacerbate disease and without blockade with anti- GM-CSF mAb to support the proposed GM-CSF-IL-33 axis. No data on blockade of endogenous IL-33 activity was presented either. These types of experiments would seem to be required to support the title.

Cromolyn was used as a well known "mast cell (MC) stabilizer, which inhibits their degranulation and cytokine release" (p13). The authors reported that the % GM-CSF+ MC (measured by intracellular FACS) was dampened by cromolyn administration as an indication of "inhibition of MC activity". How is the readout of intracellular GM-CSF related to degranulation and cytokine release?

While upregulated expression of CD131 (GM-CSF receptor β chain) is interesting the expression of the specific α binding chain is also important since the former is common to the IL-3 and IL-5 receptors. Data?

Some prior literature linking IL-33 and GM-CSF should be acknowledged.

Minor Comments

Fig. S1D Markers?

P5 etc. Remove "striking", "profound", "substantial" to describe some of the changes found in Figs. 1E and 1F. Also remove "striking" when referring to Fig.4C (p9).

P6 Reverse the order of "articular" and "intestinal", "TNF" and "IL-1 β " and "joint" and "intestinal" as they match the data presentation (Fig. S1F) better. Same comment for "intestinal" and "articular" on p8 (Fig.3F)

P8 Fig.3A. How many GM-CSF+ T cells in healthy SKG mice?

Fig.4C How accurate is the % GMPs in healthy joints?

Fig.5 What % joint cells are mast cells? Show morphology.

p13 Stz gene : "in" mast cells and not "on".

Mean arthritis score and ankle thickness (PBS) much lower in Fig.7I vs Fig 6E. Comment?

Griseri et al. manuscript, 2019.

Point by point answer to reviewers

Reviewers' comments:

We thank the reviewers for their insightful comments and suggestions, which have contributed to improve the significance of our manuscript. Our point by point answer is in plain text while the reviewers' questions appear in bold. All the changes made in the text of our manuscript have been underlined.

Reviewer #1, expert on IL-33 (Remarks to the Author):

This is a very well-presented study with convincing data addressing an important disease.

We thank the reviewer for the positive comment regarding the data presented.

That Neutrophils driven by GMCSF via myelopoiesis is important in arthritic diseases has been well-documented. GMCSF is produced by several cell types, including Mast cells. The present study links IL-33 to MCs which produce GMCSF that drives myelopoiesis that recruit neutrophils to the inflammatory joints. Infection is also an important factor. SKG mice do not develop inflammatory disease in a germ-free environment. Several questions emerge from this study.

1. How important is this axis compared to other pathways, such as CD4 cells in SpA? IL-33 was given in relatively high quantity to induce SpA (1 ug, 2x weekly for 5 weeks). Neutralisation of IL-33 or IL-33 or ST2 ko mice crossed to SKG mice might resolve this issue.

We thank the reviewer for these useful comments and have answered the several interesting points raised here:

A/ The reviewer is right to highlight that several immune pathways are likely involved in spondyloarthritis (SpA) development, i.e. innate and adaptive immune pathways (as highlighted in ref 12 and 34, referring to the human pathology). Regarding the SKG model of inflammatory arthritis, although our findings showed that mast cells contribute to SpA development, as shown by the decrease in clinical score of arthritis with treatment with the mast cell inhibitor Cromolyn (Figure 6F), we have highlighted on page 12 of the Results that ILC are also known to contribute to arthritis development in SKG mice (ref 36) and confirmed that ILC are major source of GM-CSF (Figure 5D and ref 36). As suggested by the reviewer, we performed a new experiment to investigate the role of CD4 T cells by using a potent CD4 T cell depleting antibody (**new Figure S5A**), which revealed that CD4 T cells were not absolutely required for the development of curdlan-triggered SpA in arthritis prone SKG mice (**new Figure S5A** and page 12). This result highlights the importance of innate cells in SpA development.

B/ The reviewer highlighted that a relatively high quantity of recombinant IL-33 was injected to SKG mice to aggravate disease. We now provide data showing that the injection of half of the dose of IL-33 [0.5 µg per injection, twice per week for 5 weeks] still aggravated SpA to a similar extent as with the initial dose (**new Figure S6F**).

C/ To address the comment regarding neutralization of IL-33, we performed a new set of experiments to block IL-33 in vivo by injecting a mouse ST2-Fc for the entire duration of the experiment. ST2-Fc treatment significantly decreased the arthritis score and joint swelling at week 2 (**new Figure 7K**) up until week 3.5 (**new Figure S6H**), confirming the involvement of IL-33 in disease in SKG mice. However, we observed that this protective effect was progressively lost and the decrease in arthritis was no longer significant at the final time point (**new Figure 7K**), possibly through a late compensatory effect of TNF and IL-1 as discussed on page 16 in the manuscript.

2. How do HSCs recruit neutrophils?

As clarified in our answer below, in this manuscript we do not claim that HSCs promote the recruitment of neutrophils.

In the bone marrow (BM), we showed that HSC were increased during SpA (Figure 1C) and that their differentiation was biased toward GMPs (myeloid progenitors) at the expense of erythroid (MEP) and lymphoid (CLP) progenitors (Figure 1E). GMP are progenitors for neutrophils (see schematic in **modified Figure S1D**) and their increase in SpA was accompanied by increased numbers of differentiated neutrophils in the BM of arthritic mice compared to healthy controls (Figure 1F).

In the joints, we did not report in our manuscript an increase in HSC. However we observed increased numbers of highly proliferative GMPs in the joints of inflamed SKG mice compared to controls (Figure 4C), which correlated with the emergence of a myeloid colony forming unity (CFU-GM) activity (Figure 4G) and increased numbers of differentiated neutrophils during SpA (Figure 4G). Mechanistically, we showed that GMPs isolated from the BM could migrate to the joints (Figure S4B) and that GMPs from the joints could differentiate into mature neutrophils (Figure S4B and Figure 4D). Of note, this observation is in line with the description of progenitor cells being able to migrate into peripheral tissues and differentiate locally after encountering inflammatory stimuli inside tissues (e.g. in skin, kidney, lung) see reference 27 and the newly added reference 28.

3. How relevant are the data to clinical disease in humans? The author cited studies reporting increase of IL-33 in SpA patients. In vitro studies showing that IL-33-MC-GM-CSF pathways works in human cells would also be helpful.

Our manuscript is focused on the regulation of HSC and myeloid progenitors by inflammatory stimuli, in particular the role of GM-CSF during SpA and its effect on HSPCs in the BM and at extramedullary sites. In the last figure (Figure 7) we highlighted a pathogenic IL-33/MC/GM-CSF axis in the SKG model of SpA. The in vitro studies showing that the IL-33/MC/GM-CSF axis also works in human have already been performed by several research groups and clearly showed that IL-33 can promote the secretion of GM-CSF by human MC, highlighting the relevance of the MC mediated IL-33/GM-CSF axis to inflammatory response in human. We now provide the references to two studies showing these results in human MC (references 65 and 66), and acknowledge the description of this axis in human in the Discussion on page 19.

In addition, regarding the possible role of mast cells and IL-33 in human spondyloarthritis, although the studies on this topic are relatively scarce, we highlighted in the manuscript:

- studies showing that in human inflammatory arthritis, MCs are activated in target organs and release inflammatory mediators such as IL-17A and TNF (see reference 40 and 41, and page 13).
- studies showing that IL-33 levels were substantially increased in the blood and intestine in AS patients compared to healthy controls (see reference 45 and 46, and page 14).
- studies showing that MCs are present in increased numbers in the synovial membrane of patients with SpA compared to RA patients (reference 40). We also added a new reference showing that MC were detected in the synovial fluid in a higher proportion of SpA patients compared to patients with other forms of arthritis (new reference 65). We also highlighted in the Discussion on page 20 a recent pilot study showing that targeting MC in SpA was safe and effective (reference 67) and that MCs may represent a promising and novel therapeutic target for SpA patients.

Reviewer #2, expert on cytokines in autoimmunity (Remarks to the Author):

This is an interesting report particularly the detection of GMP in inflamed joints, the putative role of GM-CSF+ mast cells and the blockade of the arthritis with anti-GM-CSF mAb.

We thank the reviewer for noting their interest in our report.

Major Comments

In general, the paper would have been strengthened by showing whether therapeutic blockade with anti-GM-CSF mAb is effective in addition to the prophylactic protocol exclusively adopted – the former is more relevant to potential clinical applications in SpA which they suggest.

This is an excellent point. To increase the relevance of our study to potential clinical applications we have now performed blockade of GM-CSF in a therapeutic setting (**new Figure 3I**). Even if the treatment was only started after the appearance of the first clinical signs of arthritis (paw and ankle swelling and redness, see individual score of arthritis in **new Figure 3I**), treatment with anti-GM-CSF was very potent at inhibiting the progression of the disease and, in most of the mice, even decreased the score of arthritis from day 9 (start of treatment) to day 28 (termination of the experiment).

Another weakness is that IL-33 was only used exogenously to exacerbate disease and without blockade with anti- GM-CSF mAb to support the proposed GM-CSF-IL-33 axis. No data on blockade of endogenous IL-33 activity was presented either. These types of experiments would seem to be required to support the title.

To strengthen the link between IL-33 and GM-CSF during SpA we performed 2 new sets of experiments in vivo.

a- In curdlan-triggered SKG mice, IL-33 was injected together with either anti-GM-CSF or an isotype control for the entire duration of the experiment (4 weeks, see **new Figure 6L**), and compared with mice injected with curdlan only. The IL-33 mediated aggravation of SpA was inhibited by GM-CSF blockade (**new Figure 6L**), as well as the IL-33 mediated increases in GMPs and neutrophils in the joints (**new Figure S6I**), strengthening our observation of an IL-33/GM-CSF pathogenic axis in SpA.

b- As detailed in the response to the question of Reviewer 1 (Question 1, point C of our answer), we have blocked IL-33 using mouse ST2-Fc injections. This decreased arthritis score and joint swelling at week 2 (**new Figure 7K**) up until week 3.5 (**new Figure S6H**), confirming the involvement of IL-33 in SpA development in SKG mice. In addition, IL-33 blockade was associated with decreased levels of GM-CSF in mast cells (**new Figure 7J**) although TNF levels were not affected in SKG mice (**new Figure S6G**).

Cromolyn was used as a well known “mast cell (MC) stabilizer, which inhibits their degranulation and cytokine release” (p13). The authors reported that the % GM-CSF+ MC (measured by intracellular FACS) was dampened by cromolyn administration as an indication of “inhibition of MC activity”. How is the readout of intracellular GM-CSF related to degranulation and cytokine release?

We previously showed in Figure 7B that IL-33 increased levels of GM-CSF released by bone marrow derived MC (BMMC). In addition, in the revised manuscript we provide FACS analysis showing that intracellular levels of GM-CSF were similarly increased in activated BMMC stimulated by IL-33 (**new Figure S6C**).

As clarified on page 13, we highlighted that mast cells (MC) release inflammatory mediators via two main distinctive pathways, i.e. degranulation of preformed compounds present in primary granules (e.g. histamine) or release of de novo synthesized cytokine (e.g. IL-6), and added a new reference reviewing these mechanisms (ref 42). We performed a new set of experiments where we observed that IL-33 did not promote BMMC degranulation (**new Figure S6B**). In addition, to strengthen our in vivo observation of cromolyn mediated decrease in SpA severity, we show in vitro that cromolyn decreased the release of GM-CSF by BMMC (**new Figure S6D**).

While upregulated expression of CD131 (GM-CSF receptor β chain) is interesting the expression of the specific α binding chain is also important since the former is common to the IL-3 and IL-5 receptors. Data?

We thank the reviewer for highlighting this point. We have now included an analysis of the expression of the *Csf2ra* gene (encoding GM-CSF receptor α chain) and showed that although *Csf2ra* was expressed by MPPs, and even at an earlier stage by the primordial HSCs, it was not upregulated during disease (**new Figure S2D**). To investigate the possible involvement of IL-3 and IL-5 in the SKG model of SpA, given these cytokines share with GM-CSF the use of GM-CSF receptor β chain for signalling, we provided a new set of in vivo experiments demonstrating that IL-3 or IL-5 blockade had no effect on disease development (**new Figure S3D**).

Some prior literature linking IL-33 and GM-CSF should be acknowledged.

We have now quoted 2 references on page 19 (references 65 and 66) showing that IL-33 can promote in vitro the release of GM-CSF by MC in human (for more details see the answer to the 3rd question of Reviewer 1), similarly to our observation with mouse BMMC and MC in the SKG model of SpA.

We have also acknowledged on page 14 prior literature demonstrating that IL-33 induces increased levels of GM-CSF in innate lymphoid cells (reference 36 in mouse, and reference 47 in human).

Minor Comments

Fig. S1D Markers?

We have added in Figure S1D a description of the FACS markers used to identify HSCs and all the downstream progenitor populations.

P5 etc. Remove “striking”, “profound”, “substantial” to describe some of the changes found in Figs. 1E and 1F. Also remove “striking” when referring to Fig.4C (p9).

We have removed those terms from the revised manuscript.

P6 Reverse the order of “articular” and “intestinal”, “TNF” and “IL-1 β ” and “joint” and “intestinal” as they match the data presentation (Fig. S1F) better. Same comment for “intestinal” and “articular” on p8 (Fig.3F)

As suggested by the reviewer, we have corrected the order of those terms to match the data presentation.

P8 Fig.3A. How many GM-CSF+ T cells in healthy SKG mice?

This information was provided in the initial manuscript and we have modified the figure title to clarify this. The right panel of **Figure 3A** represents the absolute numbers of GM-CSF⁺ CD4⁺ T cells in the joints of healthy (left column) versus arthritic (right column) SKG mice. The middle panel of **Figure 3A** depicts the percentages of GM-CSF⁺ cells among CD4 T cells in healthy (left column) versus arthritic (right column) SKG mice.

Fig.4C How accurate is the % GMPs in healthy joints?

The percentage of GMP among total cells recovered from the joint tissue in healthy mice (mean of 0.051% \pm 0.02 SD) is representative of three independent experiments as mentioned in the figure legend on page 22. This low but reproducible percentage of GMPs is consistent with the low percentage of myeloid progenitors observed in the lymph draining peripheral tissues (on average 0.03% of cells present in the lymph of steady state mice) reported in the seminal article published by the von Andrian lab (reference 27: Supplementary Figure 1). This was also confirmed by the detection of a low but consistent CFU activity from progenitors present in various peripheral tissues in steady state mice, e.g. the brain, kidney or lung (reference 27: Supplementary Figure 3).

Fig.5 What % joint cells are mast cells? Show morphology.

To be able to show MC morphology in the joints and to precisely quantify MC as requested by the reviewer, we have performed toluidine blue staining (**new Figure S5D**). This showed a ~4-fold increase in MC in the paws of spondyloarthritic mice compared to healthy mice. The morphology of toluidine blue positive cells in the joints reflects a classical MC morphology: large cells with a granular aspect (**new Figure S5D**).

p13 Stz gene : “in” mast cells and not “on”.

We have corrected this.

Mean arthritis score and ankle thickness (PBS) much lower in Fig.7I vs Fig 6E. Comment?

The difference of score between curdlan-treated SKG mice in Figure 6F (former Figure 6E) and Figure 7I is expected as the mice used in these experiments were females and males, respectively. As noted in the Material and Methods (page 24), slower and less severe SpA development in SKG males compared to females has been described previously (reference 16). We also noted in the legend of Figure 7 that the mice used were males (page 23), but to make it more clear for the reader we have now added the following sentence in the results on page 15: “To evaluate the effect of IL-33 in SpA, we treated curdlan-triggered SKG males, which develop less severe disease than females¹⁶, with twice weekly injections of IL-33 for 5 weeks (**Fig. 7C**)”.

IL-33 was potent at increasing GM-CSF production, therefore the most likely outcome of IL-33 treatment was of an increase in SpA severity. As the disease developing in SKG females is already severe and rapid, in order to help reveal a potential aggravating effect of IL-33 on SpA development we utilized SKG males in the few experiment involving in vivo injections of recombinant IL-33 (as highlighted in the legends of Figure 7C-I and 7L, and Figure S6E-F and S6I).

REVIEWERS' COMMENTS:

Reviewer #1 (Remarks to the Author):

The authors have provided careful and thoughtful response. I am content with the revision.

Reviewer #2 (Remarks to the Author):

In general, my comments and queries were addressed. Abstract should be altered to reflect some of the new data. Some minor matters should be addressed to improve further the manuscript.

Major Comments Responses

"The IL-33 mediated aggravation GM-CSF blockade (new Figure 6L)". Should be new Figure 7L?

New Figure S6D. Statistics?

2nd last line. " few experiments".

Manuscript

Fig.3I. Statistics for anti-GM-CSF vs Isotype?

p13, last line "chemokines".

p19, line 4. "Common" and reference for multiple sclerosis?

p19. "Patients" rather than "people"

p19, last line. "supporting the potential relevance to the inflammatory response to humans"

p20, line 2. Delete "in people"

p20, line 6. "trials" and use recent reviews on clinical trial data as references.

REVIEWERS' COMMENTS:

Our answer appears in bold.

Reviewer #1 (Remarks to the Author):

The authors have provided careful and thoughtful response. I am content with the revision.

We thank the reviewer for his positive evaluation of our work.

Reviewer #2 (Remarks to the Author):

In general, my comments and queries were addressed. Abstract should be altered to reflect some of the new data. Some minor matters should be addressed to improve further the manuscript.

Major Comments Responses

"The IL-33 mediated aggravation GM-CSF blockade (new Figure 6L)". Should be new Figure 7L?

New Figure S6D. Statistics?

2nd last line. " few experiments".

Manuscript

Fig.3I. Statistics for anti-GM-CSF vs Isotype?

p13, last line "chemokines".

p19, line 4. "Common" and reference for multiple sclerosis?

p19. "Patients" rather than "people"

p19, last line. "supporting the potential relevance to the inflammatory response to humans"

p20, line 2. Delete "in people"

p20, line 6. "trials" and use recent reviews on clinical trial data as references.

We thank the reviewer for these comments and have corrected accordingly the text in our revised manuscript. Of note, we have now added statistics to the Figure 3i and the Supplementary Figure 6d, and have quoted a recent article reviewing clinical trials aiming at blocking the GM-CSF inflammatory signal in arthritis (new reference 69).